# Hydroxy-octadecenoic acids instead of phorbol esters are responsible for the *Jatropha curcas* kernel cake's toxicity

Xing-Hong Wang [1,8], Jie-Qing Liu [2,8], Suiyun Chen [3,4,8], Yanfeng Yin[5], Yan Liu[1] & Changhe Zhang [6,7 ✉]

The toxic kernel cake of *Jatropha curcas* (*KCakeJ*) is an emerging health and environmental concern. Although phorbol esters are widely recognized as the major toxin of *KCakeJ*, convincing evidence is absent. Here, we show that rather than phorbol esters an isomeric mixture of 11-hydroxy-9E-octadecenoic acid, 12-hydroxy-10E-octadecenoic acid and 12-hydroxy-10Z-octadecenoic acid (hydroxy-octadecenoic acids, molecular formula $C_{18}H_{34}O_3$) is the major toxic component. The toxicities of hydroxy-octadecenoic acids on experimental animals, e.g. acute lethality, causing inflammation, pulmonary hemorrhage and thrombi, allergies, diarrhea and abortion, are consistent with those on human/animals caused by Jatropha seed and/or *KCakeJ*. The hydroxyl group and the double bond are essential for hydroxy-octadecenoic acids' toxicity. The main pathway of the toxicity mechanism includes down-regulating *UCP*3 gene expression, promoting ROS production, thus activating CD62P expression (platelet activation) and mast cell degranulation. The identification of the major toxin of *KCakeJ* lays a foundation for establishing an environmentally friendly Jatropha biofuel industry.

[1] Yunnan Institute of Microbiology, School of Life Science, Yunnan University, Kunming 650091 Yunnan, P. R. China. [2] School of Biomedical Sciences, Huaqiao University, 362021 Quanzhou, P. R. China. [3] Biocontrol Engineering Research Center of Plant Disease & Pest, Kunming 650091 Yunnan, P. R. China. [4] Biocontrol Engineering Research Center of Crop Disease & Pest, Kunming 650091 Yunnan, P. R. China. [5] Biomedical Research Center, Calmette Hospital of Kunming Medical University (Kunming First People's Hospital), Kunming 650011 Yunnan, P. R. China. [6] Center for the Research and Technology of Agro-Environmental and Biological Sciences (CITAB)/Department of Biology and Environment, Universidade de Trás-os-Montes e Alto Douro (UTAD), Apartado 1013, 5001-801 Vila Real, Portugal. [7] Instituto de Tecnologia Química e Biológica António Xavier (Green-it Unit), Universidade Nova de Lisboa, 2780-157 Oeiras, Portugal. [8] These authors contributed equally: Xing-Hong Wang, Jie-Qing Liu, Suiyun Chen. ✉email: czhang@utad.pt

*J*atropha curcas L. (hereinafter referred to as Jatropha) is a tropical and subtropical shrub or tree in the Euphorbiaceae family. Despite setbacks, Jatropha is still regarded as the most promising biofuel crop and has been widely cultivated in almost all the tropical and subtropical countries[1,2]. Approximate 100 million tons of the kernel cake of *Jatropha* (*K*Cake*J*) are projected to be generated each year worldwide, forming the major by-product of the Jatropha biodiesel industry[1].

The high protein content with a large proportion of essential amino acids indicates that *K*Cake*J* has the potential as a good protein source of feed[3]. However, due to the existence of a variety of toxins[4–6] *K*Cake*J* cannot be used as animal feed or fertilizer without effective detoxification[7]. The toxins in *K*Cake*J* are detrimental to bacteria, fungi, invertebrates, vertebrates including humans[8,9]. *K*Cake*J* also causes deterioration of soils and inhibits plant growth and seed germination[1,10]. The dispose of *K*Cake*J* without suitable detoxification is of health and eco-toxicological concern[1,4,5,7–10].

Phorbol esters are tetracyclic diterpines with a tigliane skeleton as fundamental structure. So far six phorbol diesters have been isolated from the seed oil of Jatropha[4,11] (hereinafter refer to as Jatropha phorbol esters). Jatropha phorbol esters are widely recognized as the main toxic components of *K*Cake*J*[3,8,12], although direct and convincing evidence is absent. The recognition is based on an assumption that phorbol esters are retained in *K*Cake*J* after oil extraction and on the HPLC "measurements" of phorbol esters using 12-O-tetradecanoylphorbol-13-acetate (TPA) as standard due to the unavailability of Jatropha phorbol esters[3,7,8,12]. However, the chemical structures of Jatropha phorbol esters are quite different from that of TPA. They have a macrocyclic dicarboxylic acid diester structure between the O-13 and O-16 of 12-deoxy-16-hydroxyphorbol[4,11]. TPA was first found from croton. To the best of our knowledge, no report has shown TPA's existence in Jatropha seed. Therefore, using TPA as standard to measure Jatropha phorbol esters could not give reliable result. On the other hand, Jatropha phorbol esters have very low abundance even in the seed oil and are extremely unstable[4,11]. For more than a decade, continuous efforts have been taken in our labs to isolate and purify individual phorbol esters from *K*Cake*J* for the study of the detoxification mechanism[1]. We have obtained many toxic diterpenoids from the leaves and stems of Jatropha plants[13,14]. However, neither we nor other researchers have obtained any phorbol esters from *K*Cake*J*[6]. Even though Phorbol esters are not detectable *K*Cake*J* is also highly toxic to animals and detrimental to plant seed germination and root growth[1]. We have recently developed a simple model to determine the content of the toxins of *K*Cake*J* by measuring the survival time of carp fingerlings[1]. What are the main toxic components of *K*Cake*J* remain unknown.

Hydroxy fatty acids (HFAs) widely exist in organisms. They are mainly derived from the oxidation of fatty acids. In plants, some HFAs are self-defense substances, being toxic to fungi[15]. In human and mammals, leukotoxin diol (*threo*-9,10-dihydroxyoctadec-12Z-enoic acid) and isoleukotoxin diols (*threo*-12,13-dihydroxyoctadec-9Z-enoic acid) are endogenous HFAs. They are potent cytotoxins, exerting a range of pathophysiological effects in mammals such as inhibition of mitochondrial function and increased oxidative stress[16–18].

In this report, no phorbol esters have been obtained from 2000 kg *K*Cake*J*. An isomeric mixture of 11-hydroxy-9E-octadecenoic acid, 12-hydroxy-10E-octadecenoic acid and 12-hydroxy-10Z-octadecenoic acid (hydroxy-octadecenoic acids, molecular formula of the individuals $C_{18}H_{34}O_3$) has been isolated and identified as the main toxic components of *K*Cake*J*. Experiments on carp fingerlings, mice and Guinea pigs show the key role of hydroxy-octadecenoic acids in the toxicity of Jatropha seeds and

seed cake. The underlying toxic mechanism associates with down-regulating *UCP*3 gene expression, promoting ROS production, thus activating CD62P expression (platelet activation) and mast cell degranulation. These results shed new light on both the toxicology of Jatropha seed and the impact of HFAs on animal and human health.

## Results

**Determination of the chemical structures of the toxins**. We did not obtain any phorbol esters from the 2000 kg *K*Cake*J*. Mixture 1 was isolated by subjecting the 150 kg methanol extract from 2000 kg *K*Cake*J* to repeated column chromatography, and preparatory thin-layer chromatography (TLC) combined with the carp fingering experiments to monitor and track the toxic fractions[1] during the separation process. The bulk technical grade solvents used in the extraction and separation processes were purified by distillation prior to use. Because the solvents used have a much lower boiling point than those of the individuals of Mixture 1, the Mixture 1 should have been removed from the solvents during the distillation process prior to use even though they had existed in the original commercial solvents; all the other solvents used were analytical reagents. In addition, after we obtained Mixture 1 from *K*Cake*J*, we also obtained Mixture 1 by extraction from Jatropha seed oil in some independent experiments. The extraction process from oil contained less steps than that from kernel cake and were conducted at normal laboratory scale using analytical reagents throughout the whole process. Therefore, we exclude the possibility that Mixture 1 were from the solvents. The individual components of Mixture 1 could not be separated further by the chromatographic systems that we used. The structures of the individuals of Mixture 1 were elucidated on the basis of spectroscopic methods, including high-resolution electrospray ionization mass spectrometry (HR-ESI-MS), and different NMR techniques including $^1$H-$^1$H correlation (COSY), $^{13}$C distortionless enhancement by polarization (DEPT), heteronuclear single quantum coherence (HSQC), rotating frame nuclear Overhauser effect spectroscopy (ROESY) and heteronuclear multiple quantum/bond coherence (HMQC/HMBC) NMR experiments (Supplementary Figs. 1–7).

Three hydroxyl carbon signals appeared in the DEPT $^{13}$C spectrum of mixture 1 (Fig. 1), with chemical shift of 73.31 (CH), 73.26 (CH) and 67.76 (CH), respectively. Six sp2 hybridized carbon signals with chemical shift of 133.11 (CH), 132.87 (CH), 132.55 (CH), 132.38 (CH), 132.29 (CH) and 132.02 (CH), respectively. Meanwhile three terminal fatty acid methyl carbon signals 14.13 ($CH_3$), 14.12 ($CH_3$) and 14.06 ($CH_3$) appeared in the high field. These results indicated that Mixture 1 was composed of three isomeric HFAs. $^1$H-$^1$H COSY demonstrated that the hydroxyl groups were located at the neighbor position of the double bond, and that the hydrogen (4.05–4.44) on the hydroxyl carbon was related to the hydrogen on the double bond carbon (5.38–5.68) (Supplementary Fig. 6). HR-ESI-MS (Supplementary Fig. 1) analysis showed that Mixture 1 had a unique molecular weight (Cald.: 321.2406 [M+H]$^+$ 321.2448), indicating that it was a mixture of isomers with a molecular formula $C_{18}H_{34}O_3$. Mixture 2 (Mix-2) and Compound 3 were the products obtained from the Dess–Martin oxidation reaction. A quasi-molecular ion peak [M+H]$^+$ at *m/z* 297.2418 appeared in the HR-MS/MS spectrum of Mixture 2 (Supplementary Fig. 8), and the molecular formula was concluded to be $C_{18}H_{32}O_3$ (297.2424 calculated for $C_{18}H_{33}O_3^+$). The other two fragment ion peaks, at *m/z* 279.2314 and 251.2366, were [M-OH]$^+$ and [M-COOH]$^+$, respectively (Supplementary Fig. 8).

The proton and carbon (Supplementary Figs. 9 and 10) appeared in pairs in the lower field of the NMR spectrum of Mix-

**Fig. 1 Chemical structure elucidation.** The HR-MS of Mixture 2 (Mix-2) and compound 3, as well as, the DEPT $^{13}$C spectrum of Mixture 1 (hydroxy-octadecenoic acids).

**2** and the ratio was about 1:2, indicating that the mixture contained two isomeric compounds. The HMBC spectrum (Supplementary Fig. 11) showed that the carbonyl carbon signal (202.2) was correlated with both the hydrogen signals (6.94 and 6.13) on the double bond, further confirming that the carbonyl conjugated with the double bond. Furthermore, the HR-MS/MS spectrum of Mix-2 had a base peak at $m/z$ 183.1376 and a fragment ion peak at $m/z$ 169.1220 whose mass was 14 lower than the base peak (Fig. 1 and Supplementary Fig. 8). These two peaks were the α-cleavage between the carbonyl group and the double bond, being fragment ion peaks with positive charge located on the double bond carbons. These data suggested that the two compounds in Mixture 2 were 7-carbonyl-8E-octadecoienic acid and 8-carbonyl-9E-octadecoienic acid, respectively. According to the hydrogen spectrum (Supplementary Fig. 9), the chemical shift of the hydrogen located on the carbon adjacent to the carbonyl group was 6.14 (dd, $J = 15.9$, 1.8 Hz, 1H). The coupling constant indicated that the double bond was trans. Therefore, Mixture 2 was finally determined as 12-oxo-10E-octadecenoic acid and 11-oxo-9E-octadecenoic acid. The other spectra of Mixture 2 are

shown in Supplementary Fig. 12 (HSQC), Supplementary Fig. 13 (COSY) and Supplementary Fig. 14 (ROESY).

There was an M+H peak at $m/z$ 319.2249 on the HR-ESI-MS spectrum of **3** (Supplementary Fig. 15), indicating a molecular formula $C_{18}H_{32}O_3$ (319.2249 for $C_{18}H_{32}O_3Na$). A carbonyl group (203.0) and a double bond (148.2 and 126.4) existed on the $^{13}$C-NMR spectrum (Supplementary Fig. 16). One methyl hydrogen signal (0.80, t, $J = 7.2$ Hz) and two hydrogen signals connected to the double-bonded carbon (6.12, brd, $J = 12.0$ Hz; 183.1376, m) appeared on the $^1$H-NMR spectrum (Supplementary Fig. 17). The HMBC (Supplementary Fig. 18) showed that the carbonyl carbon was correlated with both the 6.12 and 6.01–6.05 hydrogen signals, indicating that the carbonyl group was adjacent to the double bond. In addition, the coupling constant of hydrogen 6.12 was 12 Hz, showing that the double bond was cis (Z). On the HR-ESI-MS spectrum of **3**, there was a fragment ion peak at $m/z$ 183.1397 (Fig. 1). Finally, **3** was identified as 12-hydroxy-10Z-octadecenoic acid (Fig. 2). The other spectra of **3** are shown in Supplementary Fig. 19 (HSQC), Supplementary Fig. 20 ($^1$H-$^1$HCOSY) and Supplementary Fig. 21 (ROESY).

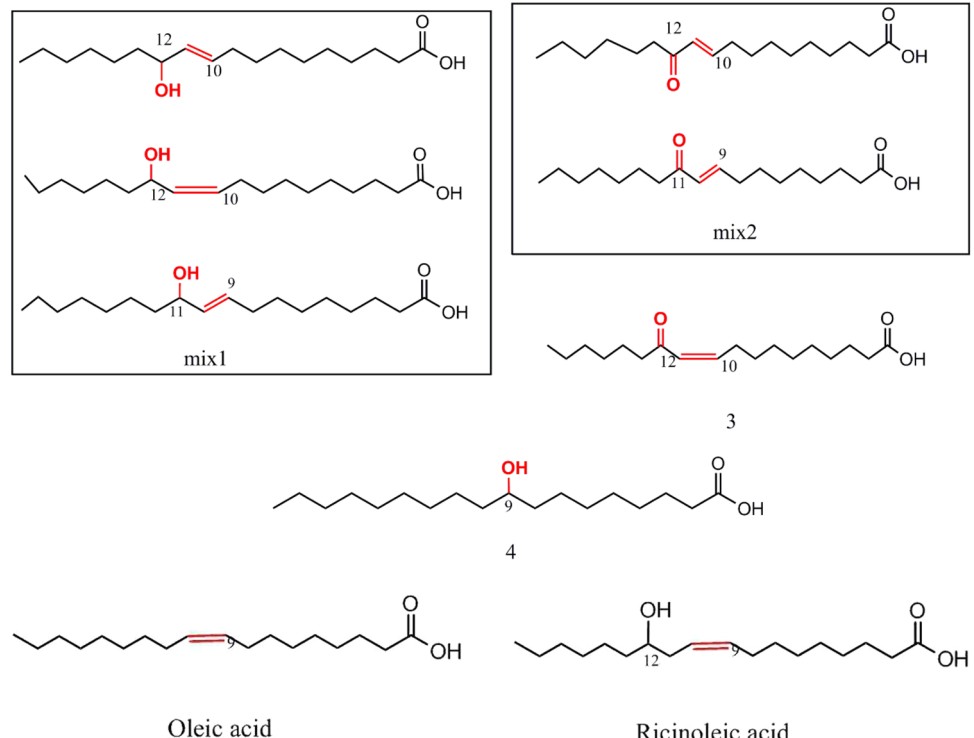

**Fig. 2 Comparison of the chemical structures of the toxins obtained with those of oleic acid and ricinoleic acid.** Chemical structures of mixture 1 (mix1, hydroxy-octadecenoic acids), Mixture 2 (mix2), compounds 3 and 4, oleic acid and ricinoleic acid. mix2 and compound **3** were the oxidized products of mix1 after the Dess-Martin oxidation.

**Table 1 The average survival time of the carp fingerlings incubated in the water containing different 18 carbon fatty acids at 50 mg l$^{-1}$.**

| Compound | Con | OA | ROA | HOEAs | Mix2 | Com3 | Com4 |
|---|---|---|---|---|---|---|---|
| Survival time (min) | 8640 ± 368[e] | 8380 ± 308[e] | 900 ± 53[c] | 45 ± 2.2[a] | 47±2.1[a] | 780 ± 28[b] | 1800 ± 76[d] |

Values are means ± STDEV ($n = 6$ independent carp fingerlings); values with different letters are significant different at $p < 0.05$. Control (Con), received the same amount of DMSO as the treated groups; ROA, ricinoleic acid (12-hydroxy-9Z-octadecenoic acid), an isomer of the toxic hydroxy-octadecenoic acids. OA, oleic acid (9Z-octadecenoic acid); HOEAs, hydroxy-octadecenoic acids, an isomeric mixture of 11-hydroxy-9E-octadecenoic acid, 12-hydroxy-10E-octadecenoic acid and 12-hydroxy-10Z-octadecenoic acid isolated from the kernel cake of Jatropha (KCakeJ); Mix2, Mixture 2 (12-oxo-10E-octadecenoic acid and 11-oxo-9E-octadecenoic acid), one part of the Dess–Martin oxidized product of hydroxy-octadecenoic acids; Com3, compound 3 (12-oxo-10Z-octadecenoic acid) the other part of the oxidized product of hydroxy-octadecenoic acids; com4, compound 4 (9-hydroxy-octadecanoic acid) isolated from KCakeJ.

Dess–Martin oxidation reaction selectively oxidizes a hydroxyl group to a carbonyl group[19]. From the structures of Mix-**2** and **3**, mixture **1** was concluded to be isomers of 12-hydroxy-10E-octadecenoic acid, 11-hydroxy-9E-octadecenoic acid and 12-hydroxy-10Z-octadecenoic acid (hereinafter referred as hydroxy-octadecenoic acids, molecular formula of the individuals C$_{18}$H$_{34}$O$_{3}$, Fig. 2). We have tried GC and HPLC methods but could not establish a linear relationship to determine the content of the individual compounds in the mixtures. It is worth mentioning that the hydroxyl oxygen of the individuals of hydroxy-octadecenoic acids is attached to the neighboring carbon of the double bond carbon. The rest compound was identified as 9-hydroxyoctadecanoic acid (compound **4**, Fig. 2).

The extracted toxic components from the seed oil were found to be basically the same as the toxic components from the KCakeJ.

**Cause acute lethality and internal organ inflammation.** As shown in Table 1, all the HFAs were lethal to the carp fingerlings. Mixture 1 (hydroxy-octadecenoic acids) killed all the carp fingerlings within 45 min. It is the most toxic component and has the highest content in the KCakeJ. Thus hydroxy-octadecenoic

acids were determined as the main toxic component of the KCakeJ. **4** is a saturated HFA (Fig. 2). **4** was toxic; while its toxicity was much weaker than that of the compounds that have both a hydroxyl group and a double bond, such as, ricinoleic acid (12-hydroxy-9Z-octadecenoic acid, Fig. 2) and hydroxy-octadecenoic acids. However, another octadecenoic acid tested, oleic acid (9Z-octadecenoic acid, Fig. 2), which does not have a hydroxyl group, was not toxic to the carp fingerlings (Table 1). These results indicate that the hydroxyl group is the structure basis for the toxicity of an HFA and that the toxicity increased in the presence of a double bond in the carbon chain; while a double bond alone does not confer toxicity to a fatty acid. The average survival time treated by hydroxy-octadecenoic acids is 20-fold shorter than that treated by ricinoleic acid (45 min vs. 900 min, Table 1). This result indicates that the toxicity of hydroxy-octadecenoic acids is much stronger than that of ricinoleic acid. Ricinoleic acid (12-hydroxy-9Z-octadecenoic acid, Fig. 2) is an isomer of hydroxy-octadecenoic acids. The hydroxyl oxygen of ricinoleic acid is attached to the carbon next to the neighboring carbon of the double bond carbon; while the hydroxyl oxygen of hydroxy-octadecenoic acids is attached to the neighboring carbon of the double bond carbon. A closer location of the hydroxyl

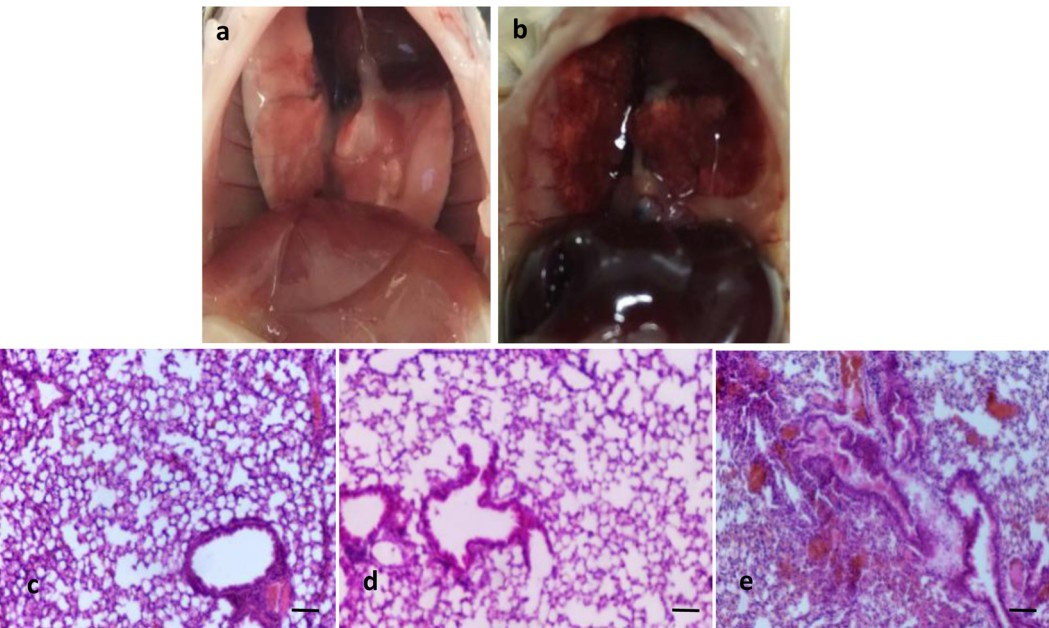

**Fig. 3 Pathological effects of tail intravenous injection of hydroxy-octadecenoic acids on mouse. a**, **b** thorax anatomic photos of the DMSO control and the hydroxy-octadecenoic acids-treated group (100 mg kg$^{-1}$ body mass), respectively. **c**–**e** Lung tissue microscopic sections (Bar, 10 μm) of normal control, DMSO control and the hydroxy-octadecenoic acids-treated group, respectively. DMSO control received the same dose of DMSO as the hydroxy-octadecenoic acids-treated group. Hydroxy-octadecenoic acids is an isomeric mixture of 11-hydroxy-9E-octadecenoic acid, 12-hydroxy-10E-octadecenoic acid and 12-hydroxy-10Z-octadecenoic acid obtained from the kernel cake of Jatropha.

group to the double bond carbon of hydroxy-octadecenoic acids confers themselves a much stronger toxicity than ricinoleic acid. These results indicate that the hydroxyl group and its relative location to the double bond play a key role in the toxicity of the unsaturated HFA.

Upon low dose tail intravenous injection of mixture 1 (hydroxy-octadecenoic acids) at 50 mg kg$^{-1}$ body mass, the mice showed twitching, mental inactive and reduced motor activity symptoms, but returned to normal state a few minutes later. These symptoms were also observed in animals fed with KCakeJ[20,21]. Continuous administration at 50 mg kg$^{-1}$ body mass day$^{-1}$ for three days, no more poisoning symptoms appeared during the following two weeks. At higher doses (≥100 mg kg$^{-1}$ body mass) the mice died immediately after administration (at 200 mg kg$^{-1}$ body mass, 6 in 6 mice died within 1 min) with a median effective lethal dose of 112.5 mg kg$^{-1}$ and a 95% confidence interval of 111.798–113.224 mg kg$^{-1}$ body mass. All of the dead mice had hemorrhage in their thoracic cavities. The liver blackened. Infiltration of numerous inflammatory cells occurred around the blood vessels and the bronchi in the administration group at 200 mg kg$^{-1}$ body mass; in the alveolar cavity there were a large number of red blood cells, showing embolization (Fig. 3). Thrombi in the lung tissue indicates that thrombosis leads to vein flow obstruction and blood spillover out of the lung surface. The alveolar structure of the mice of the DMSO control group was intact; intraperitoneal hemorrhage did not occur in this group as well. There were no macroscopic pathological differences between the organs of the DMSO control and those of the normal control group.

The lethal effects and the toxic symptoms of hydroxy-octadecenoic acids are consistent with those caused by the intake of Jatropha seeds or kernel cake of a variety of animals, such as, mouse, rat, chicken, sheep, pig, goat[20,22,23], rabbit[24], and calve[25].

**Cause skin inflammation, tail necrosis and detachment**. Inflammation appeared within 3 days after the topical application of hydroxy-octadecenoic acids onto the Guinea pig skin (Fig. 4a–l). The higher the concentration of hydroxy-octadecenoic acids resulted in the heavier inflammation and the longer healing time. As shown in Fig. 4i–l, the Guinea pig had severe pathological changes inside the skin after the application of hydroxy-octadecenoic acids. The most important change was the occurrence of a large number of inflammatory cells.

Subcutaneous injection of hydroxy-octadecenoic acids at 100 mg kg$^{-1}$ body mass resulted in visible necrosis around the injection spot of the tails of all the treated mice (Fig. 4m–p). The necrosis finally led to the detachment of the tail tips from the necrotic part (Fig. 4o, p). The experimental results are consistent with the common knowledge that KCakeJ induces skin inflammation. Consistent with our findings, the application of the toxic fraction of Jatropha seed oil to the skin of rabbits, mice or rats produced a severely irritant reaction followed by necrosis[26].

**Hydroxy-octadecenoic acids cause mice diarrhea**. Diarrhea occurred within 2–4 h after the feeding of hydroxy-octadecenoic acids at 100 mg/mouse. The symptom alleviated after food ingestion. The feces of the mice in the groups administrated at lower doses became soft. No other adverse physiological reactions were observed in the experimental groups compared with the control group, and no visible lesions were found in the internal organs. However, continuous feeding of hydroxy-octadecenoic acids the mouse kept diarrhea. Hydroxy-octadecenoic acids gavage feeding did not cause strong acute toxicity (immediate death) as intravenous injection. The soft stools and diarrhea indicated mild pathological changes in the intestines.

Diarrhea has been demonstrated as one of the major symptoms of Jatropha seed and KCakeJ. The reported victims include both children and adults[27,28], and a variety of animals, such as mice[29], rats[20,26], rabbits[26], calves[25], goats[30,31], sheep[30,32] and pigs[23]. Similar to our results, feeding KCakeJ also did not cause immediate death of mice[29], rats[20,26], rabbits[26], calves[25], goats[30,31],

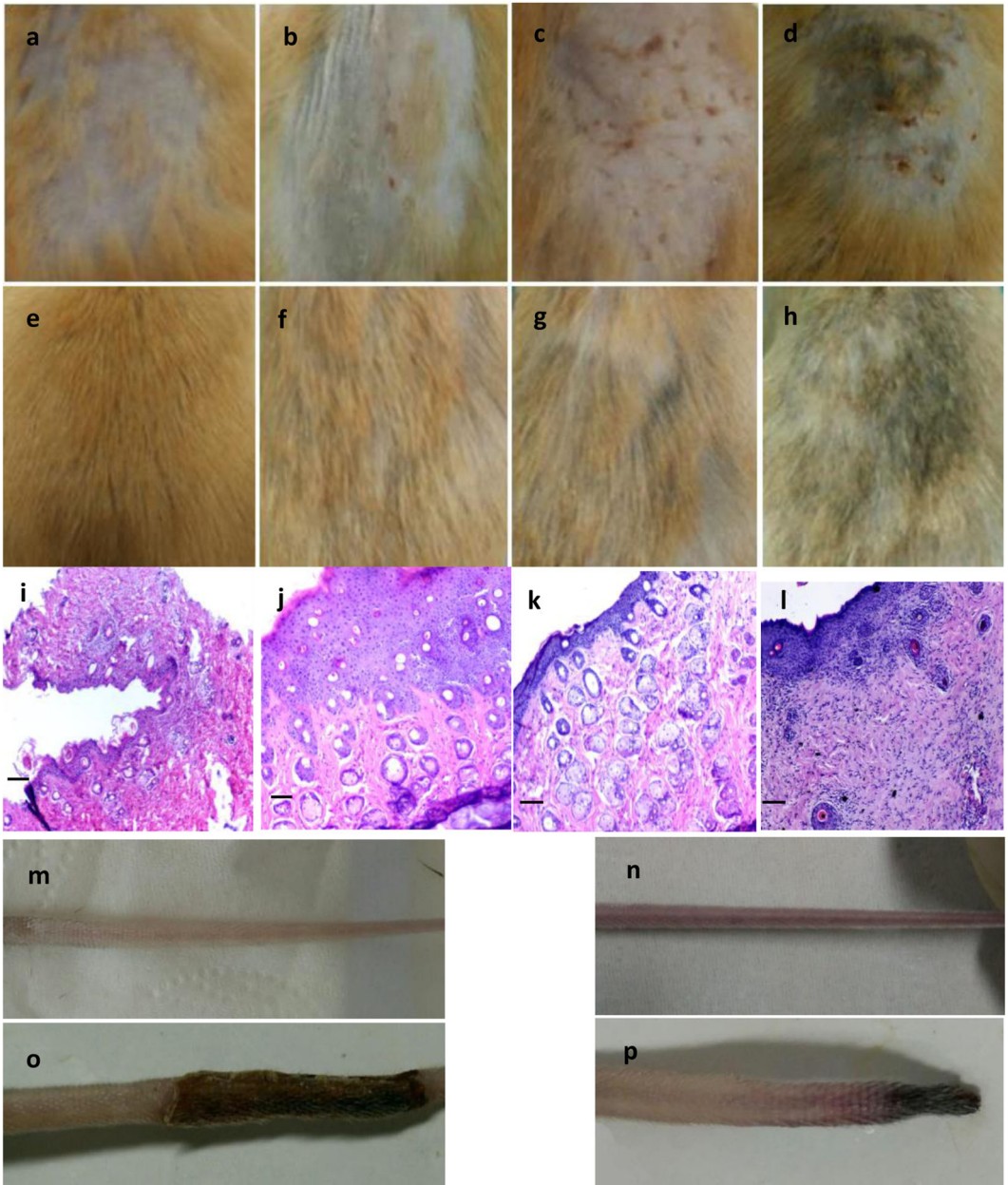

**Fig. 4 Toxic effects of the topical application of hydroxy-octadecenoic acids.** Toxic effects of topical application of hydroxy-octadecenoic acids on Guinea pig's skin inflammation (**a–i**) and the toxic effects of subcutaneous application of hydroxy-octadecenoic acids on mouse tail necrosis and tail tip detachment (**m-p**). Fifty µl DMSO containing 0, 2, 10 or 100 mg l$^{-1}$ hydroxy-octadecenoic acids was evenly applied to the surface of the shaved skin of the Guinea pigs and then covered with a piece of medical gauze and fixed by bandage tapes. The higher the concentration resulted in the heavier inflammation and the longer healing time. As shown in Fig. 4i–l, the Guinea pig had severe pathological changes inside the skin after the application of hydroxy-octadecenoic acids. A large number of inflammatory cells in the skin of the mice treated at higher concentrations of hydroxy-octadecenoic acids. **a–d** one week after application; **e–h** four weeks after application; **i–l** skin pathological sections (Bar, 10 µm). **a**, **e**, **i** control; **b**, **f**, **j** treated at 2 mg l$^{-1}$; **c**, **g**, **k** treated at 10 mg l$^{-1}$; **d**, **h**, **l** treated at 100 mg l$^{-1}$. Hydroxy-octadecenoic acids was applied to mouse tail at 100 mg kg$^{-1}$ body mass by subcutaneous injection using 30 µl DMSO as vehicle; mice in the control group only received 30 µl DMSO. **m** and **n** (control), one week and two weeks after subcutaneous injection of the same dose of DMSO alone as the hydroxy-octadecenoic acids-treated group on mouse tail, respectively; **o** and **p**, one week and two weeks after subcutaneous injection of hydroxy-octadecenoic acids at 100 mg kg$^{-1}$ body mass on mouse tail, showing the tail prior to and after tail tip detachment, respectively. Hydroxy-octadecenoic acids is an isomeric mixture of 11-hydroxy-9E-octadecenoic acid, 12-hydroxy-10E-octadecenoic acid and 12-hydroxy-10Z-octadecenoic acid obtained from the kernel cake of Jatropha.

sheep[30,32] and pigs[23]. Our work indicates the role of hydroxy-octadecenoic acids in Jatropha-seed-caused-diarrhea.

In summary, our results suggest the role of hydroxy-octadecenoic acids in the toxicity of Jatropha seed and *K*Cake*J*. The underlying mechanism is reported in the following sections.

**Down-regulate *UCP*3 expression and promote ROS production.** The uncoupling protein3 (UCP3) is located at the inner membrane of mitochondria. It regulates energy balance, basic metabolism and lipid metabolism[33,34]. UCP3 also protects cells function from oxidative stress[35]. As shown in Fig. 5a, hydroxy-octadecenoic acids significantly ($p = 0.0017$) down-regulated

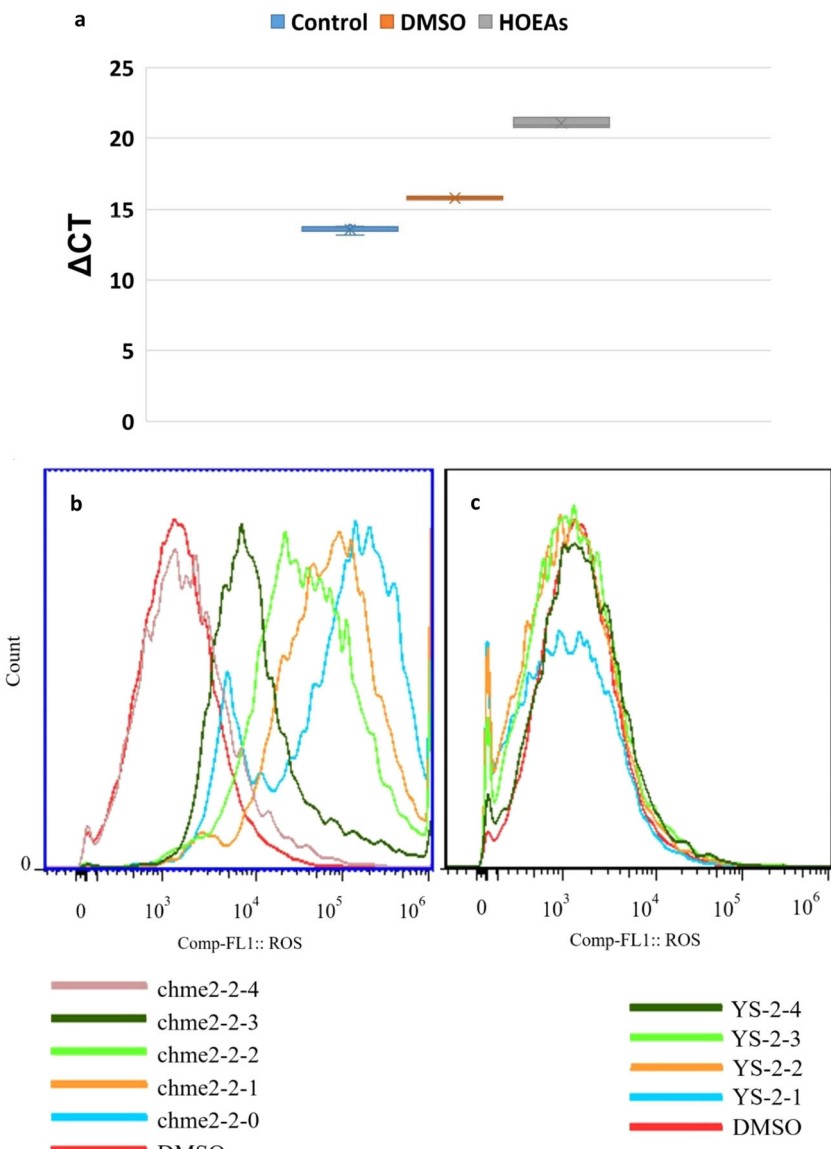

**Fig. 5 Effects of hydroxy-octadecenoic acids on *UCP3* gene expression and ROS production.** Effects of hydroxy-octadecenoic acids (HOEAs) on the expression of *UCP3* gene of L6 cells (**a**) and on the production of ROS of MCF-7 cells (**b**, **c**). $\triangle$CT = CT value of the target gene (*UCP3*)−CT value of the reference gene (β-actin gene), the box-and-whisker statistics plot was produced by Excel 2016 ($n = 3$ biologically independent samples), differences were identified using two-tailed T-Test, DMSO vs. HOEAs, $p = 0.0017$; the DMSO group received the same amount of DMSO as the HOEAs-treated group but without HOEAs (**a**); chme2-2-0, chme2-2-1, ……, and chme2-2-4, hydroxy-octadecenoic acids $2^0$-fold diluted, $2^1$-fold diluted, ……, and $2^4$-fold diluted, respectively (**b**); YS-2-1, YS-2-2, YS-2-3, YS-2-4, oleic acid $2^1$-fold, $2^2$-fold, $2^3$-fold and $2^4$-fold diluted, respectively (**c**); the concentration of the stock solutions was 100 mg l$^{-1}$. Y Axis—the number of cells (**b**, **c**), X Axis—fluorescence intensity (**b**, **c**). Hydroxy-octadecenoic acids is an isomeric mixture of 11-hydroxy-9E-octadecenoic acid, 12-hydroxy-10E-octadecenoic acid and 12-hydroxy-10Z-octadecenoic acid obtained from the kernel cake of Jatropha.

*UCP3* gene expression in L6 cells. Downregulation of UCP3 results in a decrease in muscle efficiency and/or in additional non-contractile ATP-consuming[36]. The result suggests that hydroxy-octadecenoic acids might be responsible for the weakness and fatigue of the limbs and reduced motor activity of the victims caused by the intake of *J. curcus* seeds[31] and seed cake[20]. Mitochondria are the major source of ROS, loss or inhibition of *UCP3* leads to increased reactive oxygen species (ROS) production[34,37]. Strong increases in ROS can cause contractile dysfunction and muscle atrophy, which both promote muscle weakness and fatigue[38,39]. As expected, hydroxy-octadecenoic acids significantly promoted ROS production of the MCF-7 cells with concentration dependence (Fig. 5b). However, oleic acid (9Z-octadecenoic acid) did not promote ROS formation (Fig. 5c).

These results once again indicate that the hydroxyl group is required for the toxicity of the HFAs.

ROS include superoxide anions, hydroxyl radicals, hydrogen peroxide, lipid peroxides, etc.[40,41] Normal levels of ROS play an important role in physiological activities such as signal transduction, growth, Ca$^{2+}$ signaling pathways, and regulation of redox-sensitive gene expression[41]. Excessive levels of ROS cause an imbalance between the oxidation products and the antioxidant capacity of cells, resulting in the formation of oxidative stress and various diseases. The generation of ROS is the core of many inflammatory diseases and involves more than 100 diseases, including major diseases that threaten human health, cardiovascular diseases, diabetes, neurodegenerative diseases (such as Parkinson's disease and Alzheimer's disease), acquired

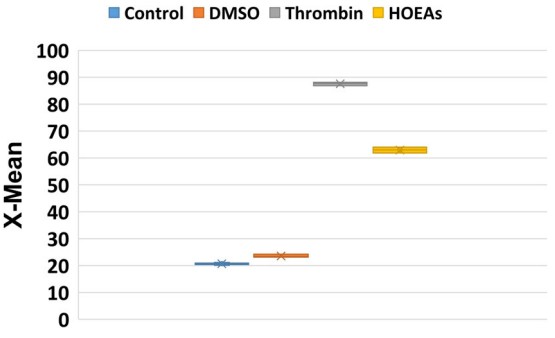

**Fig. 6 Effects of hydroxy-octadecenoic acids (HOEAs) on the expression of platelet surface antigen CD62P.** X-Mean, the mean fluorescence intensity of CD62P. The box-and-whisker statistics plot was produced by Excel 2016 ($n = 3$ biologically independent samples), differences were identified using two-tailed T-Test, DMSO vs. HOEAs, $p = 1.35E-05$, Thrombin vs. HOEAs, $p = 6.06E-05$. The DMSO group received the same amount of DMSO as the HOEAs-treated group but without HOEAs. Control was blank without any treatment. Hydroxy-octadecenoic acids is an isomeric mixture of 11-hydroxy-9E-octadecenoic acid, 12-hydroxy-10E-octadecenoic acid and 12-hydroxy-10Z-octadecenoic acid obtained from the kernel cake of Jatropha.

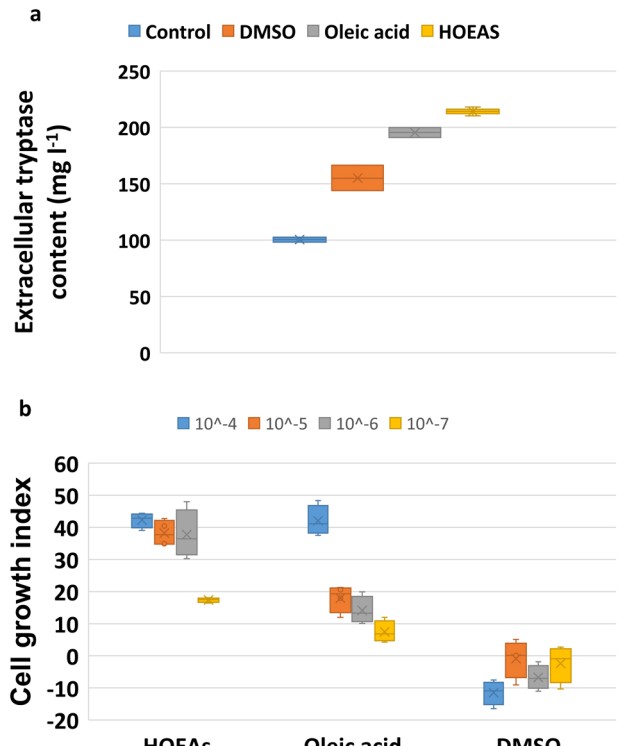

**Fig. 7 Hydroxy-octadecenoic acids (HOEAs) promote mast cell degranulation and MCF-7 breast adenocarcinoma cell growth.** Effects of hydroxy-octadecenoic acids (HOEAs) on the expression of extracellular tryptase of human LAD2 mast cells (**a**) and on the proliferation of human MCF-7 breast adenocarcinoma cells (**b**). The box-and-whisker statistics plots were produced by Excel 2016 ($n = 3$ biologically independent samples for **a**, $n = 4$ biologically independent samples for **b**); differences were identified using two-tailed T-Test, DMSO vs. HOEAs, $p = 0.013$ (**a**), DMSO vs. Oleic acid, $p = 0.01$ (**a**), HOEAs vs. Oleic acid, $p = 0.005$ (**a**). The DMSO group received the same amount of DMSO as the treated groups alone; Control was blank without any treatment. Hydroxy-octadecenoic acids is an isomeric mixture of 11-hydroxy-9E-octadecenoic acid, 12-hydroxy-10E-octadecenoic acid and 12-hydroxy-10Z-octadecenoic acid obtained from the kernel cake of Jatropha.

immunodeficiency syndrome, stroke, and aging[40–42]. ROS play a critical role in platelet activation and thrombus formation[43,44] and in mast cell degranulation[45]. In addition, ROS mediate secretory diarrhea[46]. Moreover, ROS are also related to spontaneous abortion and recurrent pregnancy loss[47]. Therefore, it is suggested that the generation of ROS triggered by the reduced *UCP*3 expression plays a key in the toxicity of hydroxy-octadecenoic acids. That oleic acid did not stimulate ROS production is consistent with its non-toxicity to the carp fingerlings as shown in Table 1.

**Activate platelets and promote blood coagulation.** Figure 6 shows that hydroxy-octadecenoic acids significantly ($p = 1.3E-05$) activated the expression of platelet CD62P. Upon activation, platelets express large amount of CD62P protein which is rapidly mobilized from α-granules to the platelet surface[48]. CD62P accumulation is one of the specific indicators that intuitively reflect the degree of platelet activation[49]. Platelets are the smallest blood component, playing a fundamental role in thrombosis and hemostasis[50,51], innate immune and in the development of inflammation[52,53]. Activated platelets trigger the activation of blood coagulation as well[54].

The mean blood coagulation time with the treatment of hydroxy-octadecenoic acids ($90 \pm 0$ s) was significantly shorter than those of the DMSO control group ($133.3 \pm 12$ s) and of the thrombin (a coagulation factor) treated group ($113.3 \pm 4.7$ s) ($p = 0.039$ and $p = 0.02$, respectively). In other words, the coagulation effect of hydroxy-octadecenoic acids is significantly stronger than thrombin. The natural blood coagulation time (blood without any treatment) was $263.3 \pm 12.5$ s. The coagulation effect of hydroxy-octadecenoic acids is associated with their impact on platelets activation. The effects of hydroxy-octadecenoic acids on inducing blood coagulation and CD62P expression (platelets activation) suggests their role in pro-inflammation and thrombosis of *K*Cake*J*. The effects of hydroxy-octadecenoic acids on platelets activation and coagulation explain the hydroxy-octadecenoic acids-caused pulmonary embolism and tail necrosis described previously and the necrosis caused by Jatropha seed oil[25].

**Stimulate mast cell degranulation and MCF-7 cell growth.** As shown in Fig. 7a, hydroxy-octadecenoic acids significantly ($p = 0.013$) increased the extracellular tryptase level of human LAD2 mast cell cultures. Tryptase is a major mediator of mast cell secretory granules and has been used as a marker for mast cell activation/degranulation[55,56]. Mast cells are widely distributed around the microvasculature of the skin and visceral mucosa. Mast cells contain secretory granules. Various bioactive molecules in the granules are released from mast cells after being stimulated, resulting in rapid allergic/inflammatory reactions in the tissues[55,57,58]. Mast cells play a key role in initiation of allergic reactions and inflammation[59–61]. Mast cell degranulation is the main pathological cause of allergies and urticaria[57–61]. That hydroxy-octadecenoic acids stimulate mast cell degranulation explains why the extract of Jatropha kernel cake and seeds caused urticaria. Hydroxy-octadecenoic acids was much stronger than oleic acid in stimulating mast cell degranulation ($p = 0.005$) (Fig. 7a), suggesting, once again, that the hydroxyl group is required for the toxicity of the hydroxy-octadecenoic acids. More recently, mast cells have been shown to modulate coagulation cascades[62].

As shown in Fig. 7b, both hydroxy-octadecenoic acids and oleic acid significantly promoted the growth of MCF-7 cancer cells. However, hydroxy-octadecenoic acids had a significantly stronger growth promotion effect than oleic acid. Hydroxy-octadecenoic acids at $10^{-7}$ mM had a similar growth promotion effect to oleic

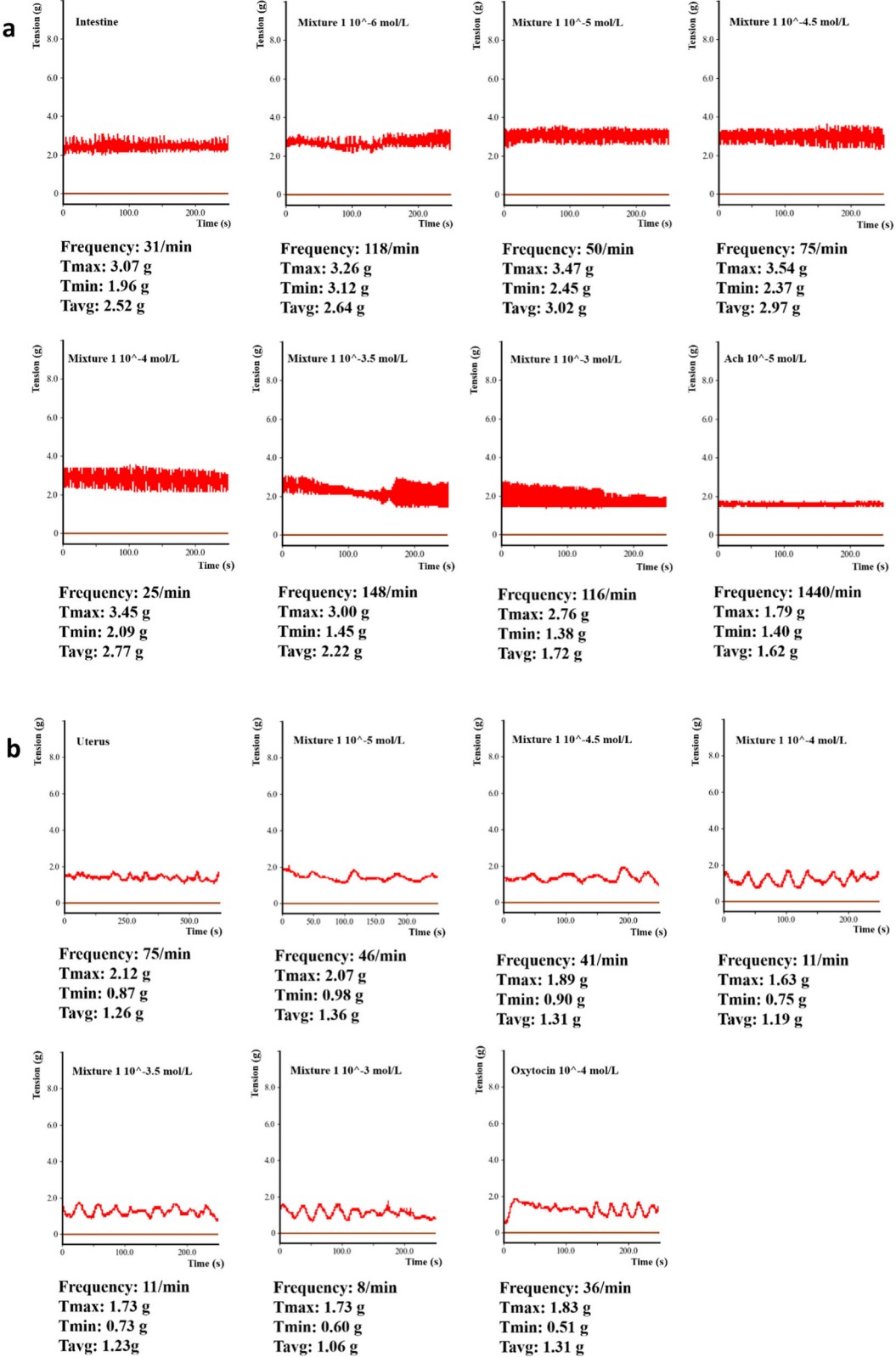

**Fig. 8 Hydroxy-octadecenoic acids disable mouse intestinal and uterine tissues.** Effects of hydroxy-octadecenoic acids on the dysfunction of mouse intestinal (**a**) and uterine (**b**) tissues. The intestinal tissue (**a**) and uterine tissue (**b**) were consecutively treated with hydroxy-octadecenoic acids at increasing concentrations indicated, and finally treated by acetylcholine (Ach) or oxytocin, respectively. Tmax, maximal tension; Tmin, minimal tension; Tavg, average tension. Hydroxy-octadecenoic acids is an isomeric mixture of 11-hydroxy-9E-octadecenoic acid, 12-hydroxy-10E-octadecenoic acid and 12-hydroxy-10Z-octadecenoic acid obtained from the kernel cake of Jatropha.

acid at $10^{-5}$ mM (growth index $17.38 \pm 0.96$ vs. $17.96 \pm 4.2$, $p = 0.8$). These results indicated that the possession of a hydroxyl group and a double bond in their molecules contributes to a stronger cancer cell growth promotion capacity to hydroxy-octadecenoic acids. Oleic acid has been proved to promote proliferation of breast cancer cells and plays an important role in breast cancer invasion and metastasis[63]. This result suggests that hydroxy-octadecenoic acids might also be carcinogenic.

**Disable mouse intestinal and uterine tissues**. The mouse intestinal contraction amplitude and frequency increased significantly with the treatment of hydroxy-octadecenoic acids at $10^{-6}$–$10^{-4}$ mM (Fig. 8a). At $10^{-3.5}$ mM, the contraction force suddenly dropped; the contraction amplitude decreased as well. Increasing the concentration to $10^{-3}$ mM, the intestine lost contraction function gradually. The addition of $10^{-5}$ mM acetylcholine (Ach) could not recover the contraction function (Fig. 8a). The average tension and contraction frequency of the mouse uterine tissue decreased upon the treatment of hydroxy-octadecenoic acids. After the consecutive treatments from $10^{-5}$ mM to $10^{-3}$ mM for 3 min at each concentration, the uterine tissue lost contraction function gradually. The addition of $10^{-4}$ mM oxytocin could not recover the uterine contraction profile to normal state (Fig. 8b). These results support the reports that intake of Jatropha seeds or *K*cake*J* causes diarrhea of human[27,28] and of different animals[20,23,25,26,29–32] and that the seeds can be used as purgatives and abortifacients in ethenomedicine[64,65]. These results also support the findings that the seeds and fruits[66] and the extract of the fruits[65] of Jatropha have abortifacient activity on rat.

Based on our findings and the reports mentioned previously, the toxicity mechanism of hydroxy-octadecenoic acids is summarized in Fig. 9.

It is ideal to quantify the relative percentage of each of the three isomers of hydroxy-octadecenoic acids. Due to lack of commercial standards of the individual isomers of the hydroxy-octadecenoic acids we were not able to establish a linear relationship to determine the content of the individual isomers by GC and HPLC methods. However, the high sample purity requirements (≥95%) of the various NMR techniques and the HR-MS technique used for the successful structure elucidation of the isomers and the fact that mixture 1 could not be separated by the chromatographic methods as mentioned previously suggest that the purity of the isomers in Mixture 1 was very high. Therefore, although we cannot exclude the existence of minor unknown impurities in hydroxy-octadecenoic acids, the impurities were not the major component of the toxins and their contribution to the toxicity was not big enough to affect our determination of the toxicity of the isomeric mixture if they did exist. The determination of the toxicity of each individual isomer was impossible due to that they are neither commercially available nor possible to be separated from hydroxy-octadecenoic acids. Hydroxy-octadecenoic acids are from Jatropha seed. Naturally, the three individual isomers of hydroxy-octadecenoic acids co-exist in the seed and kernel cake, and also exert toxicity together. The determination of the toxicity of hydroxy-octadecenoic acids on the whole mimics their natural toxicity characteristics in seed and kernel cake. Therefore, for the determination of the toxicity of hydroxy-octadecenoic acids it is not necessary to quantify the relative percentage of each of the three isomers.

## Discussion

In this work, we did not isolate any phorbol esters from the 2000 kg *K*Cake*J*. This result is consistent with our previous findings that phorbol esters were not detectable in *K*Cake*J*[1]. Pro-inflammation is one of the major toxicities of Jatropha seeds and *K*Cake*J*[26,31,67]. A recent report demonstrated that the pro-inflammatory activities of the phorbol esters from Jatropha oil were either undetectable or orders of magnitudes below the potency of TPA[68]. These results indicate that the physiological/toxicological effects of Jatropha phorbol esters are different from those of TPA and that phorbol esters are not responsible for the pro-inflammatory activities of Jatropha seeds. Together with the facts that phorbol esters found in Jatropha oil are extremely unstable[4,11] and that phorbol esters have not been proved to be at detectable level in *K*Cake*J*, it can be concluded that phorbol esters are not the major toxins of *K*Cake*J*. Our animal experiments clearly demonstrate that hydroxy-octadecenoic acids have strong pro-inflammatory activities; the underlying mechanism was found to be associated with ROS stimulated platelet activation, blood coagulation, and/or mast cell degranulation (Fig. 9). In addition, our work also demonstrates that the hydroxy-octadecenoic acids contribute to many other toxicities of Jatropha seeds and *K*Cake*J*, such as, mortality, causing coagulation and thrombosis, causing diarrhea and abortion, causing skeleton muscle weakness and lower motor activity, causing allergies and urticaria and promoting cancer cell proliferation. As summarized in Fig. 9, down-regulating *UCP*3 gene expression and thus promoting ROS production are the two fundamental events involved in the toxicities process of hydroxy-octadecenoic acids. We do not exclude the possibility that there might be other genes involved in the initiation of the toxicity. Our results provided a brief, but also clear, data chain, which indicates that *UCP*3 is relevant to the symptoms caused by Jatropha seeds/kernel cake (Fig. 9). In brief,

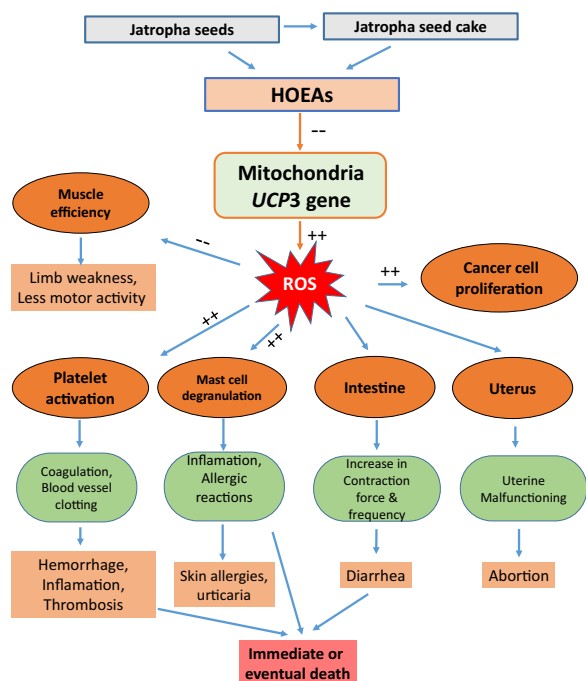

**Fig. 9 An overview of the toxicity mechanism of Jatropha seeds, kernel cake and hydroxy-octadecenoic acids (HOEAs).** Hydroxy-octadecenoic acids is an isomeric mixture of 11-hydroxy-9E-octadecenoic acid, 12-hydroxy-10E-octadecenoic acid and 12-hydroxy-10Z-octadecenoic acid obtained from the kernel cake of Jatropha. ROS short for reactive oxygen species.

the results demonstrate that the main toxins of the *KCakeJ* are hydroxy-octadecenoic acids rather than phorbol esters.

Our work, for the first time, to our knowledge, demonstrates that the hydroxy-octadecenoic acids have acute lethal effects on animals. By testing the toxicity of a serial of the 18-carbon fatty acids, including oleic acid (unsaturated fatty acids not possessing a hydroxyl group, non-toxic), **4** (saturated HFA, toxic), ricinoleic acid (unsaturated HFA, hydroxyl group attached on the carbon next to the neighbor carbon of the double bond carbon, stronger toxic) and hydroxy-octadecenoic acids (unsaturated HFA, hydroxyl group attached on the neighbor carbon of the double bond carbon, strongest toxic) (Table 1 and Figs. 5 and 7), we have demonstrated that the hydroxyl group and the double bond and their relative location play a key role in the toxicity of the fatty acids. Consistent with our findings, the hydrated products of leukotoxin (9,10-epoxy-12Z-octadecenoic acid) and isoleukotoxin (12,13Z-epoxy-9Z-octadecenoic acid), leukotoxin diol and iso-leukotoxin diol, are more toxic than the parent leukotoxin and isoleukotoxin, which suggests the role of the diols in the toxicity of the dihydroxy fatty acids[18]. The hypothetical "diol-specific" mechanism of toxicity was further confirmed by analyzing the cytotoxicity of 75 various parent olefins with different carbon chain length and their corresponding epoxides and diols[17]. The toxicity and underlying toxic mechanism of the hydroxy-octadecenoic acids deciphered in this work shed new light on our understanding of the physiology and toxicology of HFAs and their impacts on human and animal health.

The determination of the main toxins lays a basis for the study of an efficient protocol for *KCakeJ* detoxification, which is essential for upgrading Jatropha biofuel industry into a new era that not only exempts from the health and environmental concern caused by the toxic *KCakeJ* but also benefits from the added value from detoxified *KCakeJ*. Further studies are needed to develop efficient protocols to detoxify the hydroxy-octadecenoic acids of *KCakeJ* and to evaluate the effects of dietary HFAs produced from food processing and storage on human health.

## Methods

**Reagents**. All the chemicals and reagents were from Sigma-Aldrich and were Analytical Reagents except for specified elsewhere. Mixture 1 was used at 100 mg l$^{-1}$ with DMSO as vehicle (control received the same amount of vehicle without hydroxy-octadecenoic acids) in the toxicity tests, except for specified elsewhere.

**Extraction and isolation of the toxins from *KCakeJ***. The *KCakeJ* was purchased from Yunnan Shenyu New Energy Company in Kunming, Yunnan, China. The toxins in the *KCakeJ* were extracted with absolute methanol for three times. In brief, 5000 L methanol was added to an extracting tank containing 2000 kg *KCakeJ* to extract the toxins at 50 °C for three times. Previous toxicity experiments on carp fingerlings shown that methanol was an effective solvent to extract the toxins. The methanol was removed at an evaporator under vacuum at 60 °C and 150 kg extract was obtained. Thirty liters of deionized water was added to the extract and mixed homogeneously, followed by the extraction of 200 L ethyl acetate for three times in another extracting tank. Previous experiments indicate that ethyl acetate can effectively extract the toxins from the methanol extract. The water phase after the extraction was not toxic. After the removal of ethyl acetate by vacuum evaporation, fraction 2 (Fr.2) was obtained (8 kg) and confirmed to be toxic.

In the pre-experiments, we found that silica absorbed a big quantity of the sample during the chromatography progress. Every column chromatography process resulted in a big loss of the sample. Due to the difficulty to separate the toxins, the separation process included many repeated chromatography processes. Therefore, the whole separation process would lose a big quantity of sample. In addition, we hoped to obtain enough major toxins for future studies, such as, to do the toxicity tests and to do the biotransformation experiments for the detoxification studies. Furthermore, we wanted to check whether it is possible to obtain phorbol esters from an extremely big sample size. Therefore, we started the extraction and isolation process with 2000 kg *KCakeJ*.

The technical grade solvents used in the initial extraction and separation processes, including methanol (obtained from Yunnan Yun Feng Chemical Engineering Ldt), acetyl acetate (obtained from Guangxi Yi Tian Chemical Engineering Ldt) and petroleum ether (b.p. 60–90 °C, obtained from Shandong

Tian De Chemical Engineering Ldt), were distilled under 0.06 MPa to remove the impurities prior to use. During the distillation process, the top 10% and the bottom 10% of the distilled solvents were discarded; the middle 80% was collected for use. Solvents less than 10 litters were distilled in a vacuum rotary evaporator; bigger quantity of solvents were distilled in a 10 m³ steel distillation equipment. Methanol, acetyl acetate and petroleum ether were distilled at 50 °C, 42 °C and 35–40 °C, respectively.

Fr.2 was separated by silica-gel column chromatography using gradient elution of petroleum ether:acetone (100:0–5:1). Nine fractions were identified and obtained following the detection of TLC. Among them Fr.2-3, Fr.2-4 and Fr.2-9 were confirmed to be the toxic fractions using the carp fingerling model described below.

Fr.2-3 (3.73 kg, the major toxic fraction) was separated by ADS-7 column chromatography eluted with 60% and 90% methanol/water system and the toxic fraction Fr.2-3-1 (1 kg) was collected from the 90% methanol/water fraction. Fr.2-3-1 (1 kg) was separated by silica-gel column chromatography eluted with chloroform:methanol (50:1). With the detection of TCL, Fr.2-3-1-3 (Mixture 1 or hydroxy-octadecenoic acids) was obtained (100 g).

After polyamide column chromatography eluted with methanol:water (6:4–9:1), toxic fraction Fr.2-4-2 (218 g) was isolated from Fr.2-4 (469 g). Fr.2-4-2 was separated by silica-gel column chromatography by gradient elution of petroleum ether:ethyl acetate (1:0–0:1) and toxic fraction Fr.2-4-2-7 (20 g) was collected. Finally, Fr.2-4-2-7 was separated by ODS column chromatography by gradient elution of methanol:water (6:4–1:0) and compound **4** (240 mg) was obtained.

**Dess–Martin oxidation method**. 0.6 g Mixture 1 was oxidized by the Dess–Martin method. To a stirred suspension of the periodinane (1.70 g, 4.00 mmol) in CH$_2$Cl$_2$ (25 ml) under an inert atmosphere was added the Mixture 1 (0.6 g, 2.00 mmol) in CH$_2$Cl$_2$ (5 ml). After 80 min the reaction mixture was diluted with CH$_2$Cl$_2$ (60 ml) followed by a solution of NaHCO$_3$ (1.79 g) and Na$_2$S$_2$O$_3$ 5H$_2$O (5.89 g) in water (75 ml). The biphasic system was vigorously stirred until both layers became clear. The separated ether layer was washed with aqueous NaHCO$_3$ (25 ml) and brine (25 ml), dried with anhydrous Na$_2$SO$_4$ and concentrated. The resulting crude the oxidants was purified by silica-gel chromatography using gradient elution of petroleum:acetyl acetate (10:1–0:1), and the oxidized products compound **3** (459 mg) and Mixture **2** (58 mg) were obtained.

**Extraction and isolation of toxins from seed oil**. The seed oil of Jatropha was purchased from Yunnan Shenyu New Energy Company in Kunming, Yunnan, China. Five liters of seed oil was extracted with 15 L methanol–water (80:20): petroleum ether (2:1), for two times. The methanol–water phase was then extracted with ethyl acetate for three times. Eighty grams of extract was obtained from the vacuum evaporation of the ethyl acetate extract. The obtained extracted was centrifuged at 8000 rpm for 10 min. The pellet was discarded and 50 g supernatant was obtained. The supernatant was extracted with methanol–water (60:40): petroleum ether (2:1) at 3-times volume of the supernatant for three times. Eight grams of extract was obtained from the vacuum evaporation of the methanol–water phase, and separated by the methods mentioned previously. Every extraction process was evaluated by the carp fingerling model, which confirmed that the toxins were in the relevant extract obtained.

**Carp fingerling model for monitoring/testing the toxins**. The carp fingerling model[1] we established previously was used to evaluate and track the toxicity of the chromatography fractions and compounds isolated. In brief, 10 mg material/extract to be tested was dissolved in 100 μl DMSO and added to a 500-ml beaker, which contained 200 ml of edible tap water, drop by drop with agitation. In the control group, 100 μl DMSO alone was added to the 200 ml tap water in each beaker. Carp fingerlings (*Carassius auratus auratus*) were purchased from the local market and fostered in tap water for 24 h prior to the test. Three 2-cm-long carp fingerlings (about two years old) were randomly fostered in each beaker without feeding. The survival time of the carp fingerlings was recorded.

**Structure elucidation of Mixture 1**. $^1$H and $^{13}$C NMR spectra were measured on DRX-600 instruments (Bruker, Zurich, Switzerland) with Me$_4$Si (TMS) as internal standard. HRESIMS data were recorded on an API QSTAR Pulsar spectrometer. Semi-preparative HPLC was performed using an Agilent 1100 liquid chromatography with an YMC-PackProC$_{18}$ (YMC, 250 mm × 10 mm, 5 μm) column. Column chromatography was performed using silica gel (200 mesh to 300 mesh, Qingdao Marine Chemical Factory, Qingdao, People's Republic of China), Lichroprep RP-18 gel (40–63 μm, Merck, Darmstadt, Germany), or Sephadex LH-20 (Merck, Darmstadt, German). Fractions were monitored by TLC under UV light, and spots were visualized by heating silica-gel plates sprayed with 10% H$_2$SO$_4$ in EtOH. All solvents, including petroleum ether (60–90 °C), were distilled prior to use.

**Physical and spectroscopic data of isolated compounds**. Mix1 (12-hydroxy-10E-octadecenoic acid, 12-hydroxy-10Z-octadecenoic acid and 11-hydroxy-9E-octadecenoic acid): Light yellow oil. Positive HRESIMS *m/z* 321.2448 [M + Na]$^+$ (calcd for C$_{18}$H$_{32}$NaO$_3$, 321.2406). $^1$H NMR (600 MHz, MeOD): δ 5.35–5.69 (–C<u>H</u>=), 4.02–4.45 (–C<u>H</u>–OH), 1.35–2.36 (–C<u>H$_2$</u>–), 0.88 (–C<u>H$_3$</u>); $^{13}$C NMR (150

MHz, MeOD): δ 178.3 (3×C=O), 132.04–133.11 (6×-CH=), 73.31,73.26, 67.76 (3×-CH-OH), 37.3–22.6 (39×-CH$_2$–), 14.1 (3×-CH$_3$).

Mix2 (12-oxo-10E-octadecenoic acid and 11-oxo-9E-octadecenoic acid): Light yellow powder. Positive HRESIMS $m/z$ 297.2418 [M + H]$^+$ (calcd for C$_{18}$H$_{33}$O$_3$, 297.2430). $^1$H NMR (600 MHz, MeOD): δ 6.94 (1H, dd, $J$ = 14.0, 7.0 Hz, H-9 or H-10′), 6.14 (1H, dd, $J$ = 14.0, 1.5 Hz, H-10 or H-11′), 3.34 (2H, m, H-8 or H-9′), 2.61 (2H, s, H-2 or H-2′), 2.27 (2H, m, H-12 or h-13′), 1.62 (2H, m, H-3 or H-3′), 1.35 (16H, m, H-4~7 or H-4′~8′, H-14~17 or H-15′~17′), 1.26 (2H, m, H-13 or H-14′), 0.93 (2H, t, $J$ = 7.0 Hz, H-18 or H-18′); $^{13}$C NMR (150 MHz, MeOD): δ 202.3 (C-11 or C-12′), 176.3 (C-1 or C-1′), 148.4 (C-9 or C-10′), 129.6 (C-10 or C-11′), 39.5 (C-8 or C-9′), 33.5 (C-2 or C-2′), 32.1 (C-12 or C-13′), 28.9 (C-13, 14 or C-14′, 15′), 28.7 (C-15), 28.5 (C-6 or C-6′), 27.8 (C-5 or C-5′), 27.7 (C-4 or C-4′), 24.6 (C-7 or C-8′), 24.1 (C-3, C-16 or C-3′, C-16′), 22.3 (C-17 or C-17′), 13.0 (C-18 or C-18′).

Compound 3 (12-oxo-10Z-octadecenoic acid): Light yellow powder. Positive HRESIMS $m/z$ 319.2249 [M + Na]$^+$ (calcd for C$_{18}$H$_{32}$NaO$_3$, 319.2249), 297.2404 [M + H]$^+$ (calcd for C$_{18}$H$_{33}$O$_3$, 297.2430). $^1$H NMR (600 MHz, MeOD): δ 6.11–6.13 (1H, m, H-11), 6.02–6.04 (1H, m, H-10), 2.47– 2.48 (2H, m, 9-H), 2.35–2.38 (2H, m, 13-H), 2.08–2.10 (2H, m, H-15), 1.45–1.52 (4H, m, H-14, H-7), 1.31–1.36 (2H, m, H-8), 1.17–1.24 (16H, m, H-2~H-6, H-16~H-18), 0.80 (3H, t, $J$ = 6.0 Hz). $^{13}$C NMR (150 MHz, MeOD): δ 202.98 (s, C-12), 179.31 (s, C-1),148.27 (d, C-10), 126.44 (d, C-11), 43.60 (t, C-13), 36.59 (t, C-15), 31.31 (t, C-2), 29.04 (t, C-9), 28.96 (t, C-8), 25.87 and 23.85 (t, C-7 and C-14), 12.99 (q, C-18).

**Acute toxicity test on mice via tail intravenous injection**. The general aim of the in vivo animal toxicity tests was to confirm whether hydroxy-octadecenoic acids cause the same toxicity as Jatropha seeds/seed cake. All animal procedures were conducted according to conditions approved by the Committee on Animal Research and Ethics of Yunnan University; The dosage of the in vivo toxicity tests was estimated by the toxicity of hydroxy-octadecenoic acids obtained in carp fingerling tests (the same as below).

The purpose of the intravenous injection test was to observe the acute toxicity of hydroxy-octadecenoic acids. Kunming mice were randomly divided into four groups (8-week old, 18–22 g, $n$ = 6, 3 males and 3 females in each group). Each dose of Mixture 1 was dissolved in 0.03 ml DMSO and injected at 0, 50, 100 and 200 mg kg$^{-1}$ body mass, respectively.

One mouse of each group was immediately killed after the injection of hydroxy-octadecenoic acids. A lung tissue of 1.0 cm × 1.0 cm × 0.2 cm was taken from the same position of the killed mice. The tissues were sequentially subject to fixation, embedding, slicing, HE staining, decolorization, and sealing with neutral resin for microscopic observations and examinations.

**Dermal toxicity test on Guinea pig and mouse**. Male Guinea pigs were chosen for the test because of their relative large size in order to observe the toxic effects of topical application of hydroxy-octadecenoic acids on animal skin. Male Guinea pigs (about 8-week old) were randomly divided into four groups (n = 2). The hair on the back of each Guinea pig was shaved over an area of 2 cm × 3 cm. Fifty µl DMSO containing 0, 2, 10 or 100 mg l$^{-1}$ hydroxy-octadecenoic acids was evenly applied to the skin surface of the shaved zone and then covered with a piece of medical gauze and fixed by bandage tapes.

One week after the topical application of hydroxy-octadecenoic acids, one Guinea pig of each group was killed. A piece of 1.0 cm × 1.0 cm full depth skin was taken from the hair-removed area. The tissues were sequentially subject to fixation, embedding, slicing, HE staining, decolorization, and sealing with neutral resin for microscopic observations.

The purpose of the mouse tail experiment is to confirm the toxic effects of subcutaneous application of hydroxy-octadecenoic acids on mouse tail necrosis. Mixture 1 (hydroxy-octadecenoic acids) was applied to mouse tail at 100 mg kg$^{-1}$ body mass by subcutaneous injection using 30µl DMSO as vehicle; mice in the control group only received 30µl DMSO. Each group had 6 mice (3 females, 3 males).

**Acute toxicity test on mice by gavage feeding**. The purpose of this test is to check whether hydroxy-octadecenoic acids cause diarrhea. Kunming mice aged 8 weeks, weighted about 20 g each were randomly divided into 4 groups (n = 3), abstained from food but not from water for 12 h. One hundred µl Tween-80 solution containing 100, 50, 10 or 0 mg l$^{-1}$ hydroxy-octadecenoic acids was fed into the stomach of each mouse through a gavage feeding needle (4.5 cm long, 1 mm diameter). The mice were abstained from food for another 4 h. The respiration, changes in hair and eyes, intake amounts of water and food, urination and excrement were recorded. The mice were killed by cervical dislocation 12 h later. The bodies were dissected for observing the internal organs.

**In vitro toxicity tests on smooth muscle tension**. The ileum or uterine horn was cut into 1.5–2 cm long specimens. The ileum content was gently flushed out with a Pasteur pipette. One end of the ileum or uterine horn was fixed on an L-hook with suture. The other end was connected with a long suture and tied to a transducer. The other end of the transducer was connected to a balance recorder through the adjustment box. The specimen was immersed in a tube containing 10 ml of

Tyrode's solution at 37 ± 0.5 °C. Oxygen was slowly supplied to the Tyrode's solution bath. After 30 min of adaptation, the recorder was switched on. The normal activity curve of the specimen was recorded. Then the specimen was treated with different chemicals and the corresponding reactions were observed and electronically recorded.

**Rat blood coagulation test**. Female SD rats, aged 7–8 weeks, weighted 180–220 g, were used for collecting blood. Five hundred µl blood was taken from the rat heart and added to a centrifuge tube containing 500 µl mixture 1 (hydroxy-octadecenoic acids) solution (200 mg l$^{-1}$). Mixing immediately and the blood clotting time was recorded. Control only received the same amount of DMSO as the treatment.

**Cultivation of human MCF-7 cells and rat L6 cells**. The MCF-7 (ATCC_HTB22TM), LAD2 and L6 cell lines were purchased from Beina Biology, Beijing. Authentication of all the cell lines used in this study was performed by the manufacture. All the cell lines were tested negative for mycoplasma contamination. Cells were cultured in appropriate medium as described in the ATCC protocol with the addition of 10% FBS. Cell viability was determined by MTT Assay. Growth index = (OD value of the Mixture 1-treated group−OD value of the control group) /OD value of the control group × 100%.

**Determination of rat platelet CD62P activity**. Blood samples were collected and put into 2-ml centrifuge tubes containing heparin sodium from mice via retro-orbital blood collection method. The whole blood was centrifuged at 200$g$ for 10 min at room temperature to obtain platelet-rich plasma. The platelets were isolated by further centrifugation at 1200$g$ for 10 min and washed twice with PBS. Platelets were then re-suspended in PBS and transferred to 4 groups of tubes, which contained 5 µl PBS (control), DMSO solution at the same level as in the Mixture 1 group (DMSO control), hydroxy-octadecenoic acids solution, or thrombin (Melon Bio, positive control), respectively, at 1 × 10$^5$ (50 µl)$^{-1}$ each tube and incubated for 5 min at 22 ± 1 °C, then 10 µl anti-CD62P antibody (purchased from eBioscience, Lot Number 4318352; validation was performed by the manufacture.) were added, giving the final concentration of the antibody to 5 µg ml$^{-1}$, and incubated for 30 min in the dark at 22 ± 1 °C.

The platelet population was identified and gated by their characteristic forward (FSC) and side angle scatter (SSC) properties in a flow cytometer (FC 500 MCL, Beckman Coulter). Anti-CD62P antibody was used as a marker. Mean fluorescence intensity was analyzed (Supplementary Fig. 22). The amount of cells detected was about 10,000. Gating technology: FSC/SSC was the most concentrated. Cells with similar diameter and particle size accounted for about 90% of all the collected cells. The others were cell debris and miscellaneous cells. Platelets were activated upon external stimulation. The activation degree was different in relation to different treatments. Double peak did not appear in the FACS image. The data used were the average fluorescence intensity.

**Mitochondrial uncoupling protein 3 gene (UCP3) expression**. Two ml cell suspension (5 × 10$^4$ ml$^{-1}$) was transferred to each well of 6-well plates and cultured in an incubator at 5% CO$_2$ at 37 ºC. 24 h later, the medium was replaced by a complete medium that contained 100 mg l$^{-1}$ hydroxy-octadecenoic acids. Seventy-two hours later, cells were collected and RNA was extracted according to the procedures described on the promega RNA Extraction Kit. The reverse transcription was carried out according to promega reverse transcription kit protocol. The primer sequences of UCP3 and the reference gene β-actin were synthesized by Invitrogen Trading (Shanghai) Co., Ltd. according to the method described in the reference[69]. UCP3 upstream primer 5′-GGAGCCATGGCAGTGACCTGT-3′, downstream primer 5′-TGTGATGTTGGGCCAAGTCCC-3′, product 179 bp; β-actin upstream primer 5′-TGGTGGGTATGGGTCAGAAGGACTC-3′, downstream primer 5′-CATGGCTGGGGTGTTGAAGGTCTCA-3′, product 266 bp. Quantitative real time RT-PCR conditions: 1 cycle: pre-denaturation 95 °C/5 min, 40 cycles: denaturation 95 °C/15s, annealing 60 °C/1 min, extension 72 °C/30 s, 1 cycle: 72 °C/10 min. △CT = CT (cycle threshold) value of the target gene (UCP3) - CT value of the reference gene (β-actin gene).

**Human LAD2 mast cell degranulation study**. LAD2 cells were collected and washed twice with PBS. The cell pellets were re-suspended with RPMI 1640 Medium supplemented with 10% Fetal Bovine Serum alone (control) or in combination with 100 mg l$^{-1}$ mixture1 (hydroxy-octadecenoic acids) + 100 mg l$^{-1}$ DMSO (hydroxy-octadecenoic acids-treated group), with 100 mg l$^{-1}$ oleic acid + 100 mg l$^{-1}$ DMSO (oleic acid-treated group) or with 100 mg l$^{-1}$ DMSO (DMSO control) and inoculated into 48-well plates at 1 × 10$^6$ cells well$^{-1}$. Each group had triplicated wells. DMSO was used as a vehicle to dissolve mixture1 and oleic acid. The cells were cultured in a 5% CO$_2$ incubator for 5 h at 37 °C, and then the medium was collected for the analysis of extracellular tryptase content. Tryptase content was determined by the procedures provided by Millipore mast cell degranulation assay kit (Cat. No. IMM001) using spectrophotometry at 405 nm. Tryptase is an indicator of mast cell degranulation[55,56].

**Measurements of ROS in MCF-7 cells**. Mixture 1 and oleic acid solutions were prepared at 100 mg l$^{-1}$ and serially diluted by 2, 4, 8, and 16 times with complete medium. When the MCF-7 cells covered 50–60% of the well surface, the medium was replaced with 0.1 ml of the hydroxy-octadecenoic acids or oleic acid solution. The cells were incubated for 24 h. Hydroxy-octadecenoic acids and oleic acid solutions were removed. The ROS determination reagent was added at 20 μl well$^{-1}$; the cell cultures were kept in the incubator for 45 min. Then the cells were digested and suspended with 0.1 ml PBS. The cell suspension was centrifuged at 1400 rpm min$^{-1}$ for 5 min; the cell pellet was washed twice. 300 μl of PBS was added to suspend the pellet, and the fluorescence signal of the second channel was detected by the flow cytometer mentioned previously. Cell abundance: the amount of cells detected was about 10,000. Gating technology: FSC/SSC was the most concentrated. Cells with similar diameter and particle size accounted for about 75% of all the population. The others were cell debris and cells at a different growth stage. Double peak did not appear in the FACS image. The data used were the average fluorescence intensity (Supplementary Fig. 23).

**Statistics and reproducibility**. Statistics were carried out by tow-tailed T-Test: two samples assuming unequal variances in Microsoft Excel 2016. Results were expressed as mean ± standard deviation ($n \geq 3$ independent animals or biologically independent samples), and differences were considered significant at $p < 0.05$. The relevant $p$ values are shown in Results.

**Reporting summary**. Further information on research design is available in the Nature Research Reporting Summary linked to this article.

## Data availability

The authors declare that the main data supporting the findings of this study are available within the article and its Supplementary Information files. Source data for Figs. 5a and 6–7 are available in Supplementary Data. Extra data are available from the corresponding author upon reasonable request.

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

## Acknowledgements

This work was financed by two Grants from Natural Science Foundation of China (No. 21366033 and No. 21772047), South and Southeast Asia Cooperation Base on Micro-biological Resource Prevention and Utilization (2018IA100), and funded by Portugal 2020 (project number 006958; reference POCI-01-0145-FEDER-006958) under the Operational Competitiveness and Internationalization Programme and the EU's European Regional Development Fund, and Portugal national funds FCT—Portuguese Foundation for Science and Technology, under the projects FCOMP-01-0124-0124-FEDER-022692, POCI-01-0145-FEDER-006958, and UID/AGR/04033/2019, and supported by Jaime Cardoso e Freire LDA, Portugal.

## Author contributions

X.H.W. conceived the work and supervised most of the experiments, C.H.Z. analyzed the dada, interpreted the experiment results, wrote the paper, and participated in experiment design, J.Q.L. executed chemical structure elucidation, S.Y.C. participated in experiment design, Y.F.Y. carried out the experiments on toxicity tests, Y.L. carried out the experiments on toxin isolation.

## Competing interests

The authors declare no competing interests.
