## [Peer Review File · Communications Biology]

Reviewers' comments:

Reviewer #1 (Remarks to the Author):

This paper reported Hydroxy-octadecenoic acids are major toxic compounds in *Jatropha curcas* kernel cake. The work is generally logical, however, the data are too preliminary to draw the conclusion. First of all, identification of major compounds in mixture 1 and 2 should be performed and absolute configuration of compound 3 and 4 should be provided. Although there is lacking of toxicological data of isolated component/comopunds, results of toxic assays are too simple. In the current form, I suggest the publication of this work on a more specialized journal. Some questions are listed as follows:

1. In Table 1, just survival time of different compounds were provided. LD50 should be better parameters to evaluate lethal dose.
2. Fig.2-3 are phenotypical results of animal experiments. More biomarkers in serum plasma should be measured.
3. Only expression of UCP3 is not convincing to illustrate the mechanism of toxicity.
4. There are some mistakes in the manuscript that need to be revised before publication.

Reviewer #2 (Remarks to the Author):

The authors of the manuscript describe the isolation, structural identification and toxicological evaluation of metabolites isolated from the extract of kernel cake of *Jatropha curcas*. The authors claim that the isolated metabolites, consisting of an isomeric mixture of hydroxy-octadecenoic acid, are responsible for the well-known toxic effect of the kernel cake of *Jatropha curcas*. The elucidation of the structures of the metabolites has been mainly performed with the aid of spectroscopic techniques with additional synthetic modification involved. The initial toxicological evaluation is enriched by in vivo toxicological assays and elucidation of the possible cellular mechanisms of toxicity. The topic results of interest to the educated audience due to the predominant importance of *Jatropha curcas* as a source of biodiesel fuel.

The manuscript is well organized and exhaustive experiments have been performed for the determination of different toxicological profile of the extracts. At the same time the manuscript, as it is presented, is characterized by important technical and conceptual flaws that undermine its scientific value and credibility.

The following issues have been identified:

1) the structural elucidation of the identified compounds is based on the High Resolution Mass Spectroscopy (HR-MS) and both mono-dimensional and bidimensional NMR analyses of the isolated "mixture 1" and of the synthetically derived "mixture 2" and "compound 3". The manuscript does not present a detailed description of the analysis of the reported spectra. The description is limited to the identification of significant peaks that determine the presence of functional groups but do not univocally identify their relative connections in the molecule (i.e. alkylic chains).

Moreover, this analysis cannot be deducted by the reader from the spectra attached in the Supporting Information. The considered spectra are in most of the cases not readable and they miss important portions. Due to its importance, for example, is unacceptable that the ¹H-NMR of "mixture 1" is cut at 7.5 ppm. In fact, the presence of carboxylic acid in the molecule has to be confirmed by the presence of proton signal of the carboxylic acid that usually results in the spectra at lower field. For the same reason ¹³C-NMR spectra have to be reported completely with ppm range 0-220 both in monodimensional and bidimensional analysis.

In addition to this, the spectra presented in the supporting info are not always consistent with the list of the peaks reported in chemistry experimental section. For example, the carbon spectrum reported for "compound 3" in supporting information file is missing a carbonyl peak that is listed in the

chemistry experimental section.

The HR-MS spectra are not reported.

For this reason, as a reviewer, I cannot univocally determine the structure of the isolated compounds as the ones reported by the authors of the manuscript. The authors need to resolve these inaccuracies for the eventual publication of the manuscript. It would be ideal to have spectroscopic analyses of standard fatty acid present in the mixture (either from commercial source or synthesis) to make a comparison with the results obtained by the authors.

2) The manuscript does not present any indication of the relative purity of the compounds in the mixtures. This is an important conceptual flaw that undermines all the toxicological experiments performed by the authors. In fact, the presence of impurities not related to the compounds described (also in minimal percentage) could be eventually responsible for the toxic effect of the mixture. For this reason, a quantitative determination of the relative presence of single isomers in the mixture must be performed to sustain all the subsequent toxicological work.

3) The protocol for Dess-Martin oxidation of the mixture 1 is not reported anywhere in the manuscript, so the reaction cannot be replicated by the reader.

4) The toxicology experiment section is characterized by poor experimental design. For example, the doses and regimen used for the administration of the extracts in in vivo experiments lacks of rational explanation.

The paragraph in which is described the incident occurred to the researcher (spillage of the extract on the skin) does not add any scientific information on the manuscript and need to be deleted.

In different part of the manuscript hypothesis are advanced by the authors with no scientific evidence of it (for example, reasons for no toxicity after HOEAs gavage feeding). This is better to be avoided.

5) The author tried to determine a structural-toxicology relationship along all the manuscript referring to single functional groups that constitute the molecules. From the paper results that having a hydroxyl group is giving a worst toxicological profile to the molecules. As it is presented, this is conceptually wrong, since is the not just the presence of a single functional group responsible for the biological activity of the molecule.

In conclusion the manuscript as it is presented presents several issues that need to be solved by the authors for publication

Reviewer #3 (Remarks to the Author):

Reviewing the manuscript entitled "Hydroxy-octadecenoic acids not phorbol esters are responsible for the toxicity of *Jatropha curcas* kernel cake".

Overall recommendation: The manuscript investigate the toxic effects of phorbol esters (PEs) of *Jatropha curcas* (KCakeJ) using different animal models and cell lines. The toxic effects were well demonstrated and the presented results and the conclusion were discussed elegantly, that make it acceptable for publication. However, further additions and modifications should be considered in order to improve the manuscript quality before publication.

- Figures,

1- One of NMR figures should be moved to the main text as NMR spectroscopy is mainly used for identification and structural elucidation

2- The quality of figure 4 mainly Figure 4B and Figure 4C should be improved

3- The text within Figure 7 is not readable. Thus the front size should be improved

- In the conclusion authors should not exaggerate the finding page 27 line 449 "This work marks a revolutionary advance". Instead it is better to say the results..suggest , propose, show

- The samples scale of the study is very large (Kg of samples Liters of solvents) which is not usually of laboratory studies where sample size is usually mg and ml of solvents. Authors should comment in this and explain why they used such large volume of sample and how sure they are from the purity of the solvent with this large solvent. Is it possible that solvent impurities may concentrated and interfere with the results?

- Is the detected compounds are commercially available, is it possible to used standard samples of these compounds to compare the NMR spectra of standard sample with those separated from *Jatropha curcas*

Finally this work is import and presents a valuable information that may help to understand the toxic roles of isomeric mixture of 12-hydroxy-10E-octadecenoic acid, 11-hydroxy-9E-octadecenoic acid and 12-hydroxy-10Z-octadecenoic acid (HOEAs)

MS. Ref. No.: **COMMSBIO-19-1307-T**

MS. Title: Hydroxy-octadecenoic acids not phorbol esters are responsible for the toxicity of *Jatropha curcas* kernel cake

Responses to the Reviewers

Referee expertise:

Referee #1: biomass valorisation

Referee #2: chemical biologist

Referee #3: material characterization, biochemistry

Reviewers' comments:

Reviewer #1 (Remarks to the Author):

This paper reported Hydroxy-octadecenoic acids are major toxic compounds in *Jatropha curcas* kernel cake. The work is generally logical, however, the data are too preliminary to draw the conclusion.

Response: We thank you very much for your critical review and insightful comments on our MS. We hope our following answers will remove your concerns.

First of all, identification of major compounds in mixture 1 and 2 should be performed and absolute configuration of compound 3 and 4 should be provided.

Response: The identification of the major compounds in mixtures 1 and 2 is described in detail in the MS from Lines 116 to 154. The configuration of the double bond in 3 was identified as Z (cis), as described from Line 162 to 163 (highlighted in grey). The

configuration of the 9-hydroxyl in 4 was undefined for it is not the major toxic compound of the kernel cake: very low abundance, as shown in Page 31-32, (highlighted in grey), and low toxicity, as shown in Table 1.

Although there is lacking of toxicological data of isolated component/compounds, results of toxic assays are too simple. In the current form, I suggest the publication of this work on a more specialized journal. Some questions are listed as follows:

1. In Table 1, just survival time of different compounds were provided. LD50 should be better parameters to evaluate lethal dose.

Response: We agree with you that LD50 should be better parameters to evaluate lethal dose. In Table 1 we intended to compare the relative toxicity of the different C18 fatty acids at the same concentration (50 mg/l), rather than to evaluate the lethal dose. The lethal dose was reported on page 11: "At higher doses (≥ 100 mg/kg) the mice died immediately after administration with a median effective lethal dose of 112.5 mg/Kg and a 95% confidence interval of 111.798 - 113.224 mg/Kg." (Page 12, highlighted in grey in the revised MS). Comparing with the traditional LD50 method, the survival time method used much less animals and materials. In particular, it can directly compare the relative toxicity of different toxins **at the same concentration**.

2. Fig.2-3 are phenotypical results of animal experiments. More biomarkers in serum plasma should be measured.

Response: The purpose of our toxicity tests was to confirm whether the toxicity of the Hydroxy-octadecenoic acids (HOEAs) isolated from the kernel cake is consistent with the typical symptoms/toxicities observed in the animals consumed Jatropha seeds and/or seed cake described in literatures in order to address that HOEAs rather than phorbol esters are responsible for the toxicity of Jatropha seeds/seed cake. We are afraid that extra toxicological data might dilute the theme and make the MS too big to be published as one article (it already has 10.023 words and 9 figures). Although we did not measure

the biomarkers in serum, we have a Section entitled “**HOEAs activate platelets and promote blood coagulation**” (Page 20-21), which is more relevant for explaining the observed typical symptoms of Jatropha seed and HOEAs on embolism and tissue necrosis.

3. Only expression of UCP3 is not convincing to illustrate the mechanism of toxicity.

Response: As summarized in Fig 8 (Fig. 9, in the revised MS), *UCP3* gene expression resulted in ROS production, and then resulted in the downstream reactions and the symptoms caused by Jatropha seed. We understand your concern. We do not exclude the possibility that there might be other genes involved in the initiation of the toxicity. Our results provided a brief, but also clear, data chain, which indicates that *UCP3* is relevant to the symptoms caused by Jatropha seeds/kernel cake (Fig. 9). (the last two sentences have been included in Discussion and highlighted in green on Page 27). As mentioned previously, the purpose of this paper is to confirm that the HOEAs rather than phorbol esters are the major toxins of Jatropha kernel cake. The other aspects of the toxicity are not so relevant to the purpose (please refer to our answer to your Question 2).

4. There are some mistakes in the manuscript that need to be revised before publication.

Response: We have corrected all the mistakes we detected and marked them in yellow, such as “tropical application” to “topical application”, “disortionless enhancement” to “distortionless enhancement”, “silicon gel” to “silica gel”, “In brief” to “In summary”, in the revised MS.

Reviewer #2 (Remarks to the Author):

The authors of the manuscript describe the isolation, structural identification and toxicological evaluation of metabolites isolated from the extract of kernel cake of *Jatropha curcas*. The authors claim that the isolated metabolites, consisting of an isomeric mixture of hydroxy-octadecenoic acid, are responsible for the well-known toxic effect of the kernel cake of *Jatropha curcas*. The elucidation of the structures of the metabolites has been

mainly performed with the aid of spectroscopic techniques with additional synthetic modification involved. The initial toxicological evaluation is enriched by in vivo toxicological assays and elucidation of the possible cellular mechanisms of toxicity. The topic results of interest to the educated audience due to the predominant importance of *Jatropha curcas* as a source of biodiesel fuel.

The manuscript is well organized and exhaustive experiments have been performed for the determination of different toxicological profile of the extracts. At the same time the manuscript, as it is presented, is characterized by important technical and conceptual flaws that undermine its scientific value and credibility.

Response: We thank you very much for your critical review, insightful comments and helpful suggestions on how to improve our MS. We hope our answers and the newly added data and modifications will remove your concerns.

The following issues have been identified:

1) the structural elucidation of the identified compounds is based on the High Resolution Mass Spectroscopy (HR-MS) and both mono-dimensional and bidimensional NMR analyses of the isolated “mixture 1” and of the synthetically derived “mixture 2” and “compound 3”. The manuscript does not present a detailed description of the analysis of the reported spectra. The description is limited to the identification of significant peaks that determine the presence of functional groups but do not univocally identify their relative connections in the molecule (i.e. alkylic chains).

Response: Their relative connections in the molecule were determined mainly by HR-MS; we have added a figure to illustrate them in the revised manuscript, entitled as “**Fig. 1 The HR-MS of Mixture 2 (Mix-2) and compound 3, as well as, the DEPT ¹³C spectrum of Mixture 1**”, on Page 7.

Moreover, this analysis cannot be deducted by the reader from the spectra attached in the Supporting Information. The considered spectra are in most of the cases not readable and they miss important portions. Due to its importance, for example, is unacceptable

that the $^1\text{H-NMR}$ of “mixture 1” is cut at 7.5 ppm. In fact, the presence of carboxylic acid in the molecule has to be confirmed by the presence of proton signal of the carboxylic acid that usually results in the spectra at lower field. For the same reason $^{13}\text{C-NMR}$ spectra have to be reported completely with ppm range 0-220 both in monodimensional and bidimensional analysis.

Response: Thanks a lot for your advice, we have replaced all the images with those that have ppm range 0-15 in $^1\text{H-NMR}$ and 0-220 in $^{13}\text{C-NMR}$ in the revised Supplementary Information 1 (SI1).

In addition to this, the spectra presented in the supporting info are not always consistent with the list of the peaks reported in chemistry experimental section. For example, the carbon spectrum reported for “compound 3” in supporting information file is missing a carbonyl peak that is listed in the chemistry experimental section.

Response: Sometimes, it would be missing the carbon signal of carboxylic acid, however, we can see the correlations between 2.09 with COOH (179.9), 25.9 and 29.1, which indicated the carbon signal of COOH at 179.9 ppm.

The HR-MS spectra are not reported.

Response: We have included the HR-MS spectra in the revised SI1.

For this reason, as a reviewer, I cannot univocally determine the structure of the isolated compounds as the ones reported by the authors of the manuscript. The authors need to resolve these inaccuracies for the eventual publication of the manuscript. It would be ideal to have spectroscopic analyses of standard fatty acid present in the mixture (either from commercial source or synthesis) to make a comparison with the results obtained by the authors.

Response: We hope our newly included data and explanations have removed your concern on the structure elucidation.

We agree with you that “it would be ideal to have spectroscopic analyses of standard fatty acid present in the mixture (either from commercial source or synthesis) to make a comparison with the results obtained by the authors”. However, it is impossible for us to obtain a mixture that contains the same composition of standard hydroxy fatty acids as the HOEAs due to it is impossible to buy or to synthesize them either in the form of the same mixture or as individuals. We have tried many times but could not separate the individuals by HPLC and other chromatography methods (please also refer to our answer to your following questions). Thus the content of the individual hydroxyl fatty acid in the isomeric mixtures is still unknown. The standard fatty acids are not commercially available; synthesis will give a mixture of many kinds of isomers rather than a unique configuration of each individuals.

2) The manuscript does not present any indication of the relative purity of the compounds in the mixtures. This is an important conceptual flaw that undermines all the toxicological experiments performed by the authors. In fact, the presence of impurities not related to the compounds described (also in minimal percentage) could be eventually responsible for the toxic effect of the mixture. For this reason, a quantitative determination of the relative presence of single isomers in the mixture must be performed to sustain all the subsequent toxicological work.

Response: We agree with you that it will be ideal to determine the relative purity of the compounds and the relative content of each individual in the mixtures. However, we believed that to do these has technical difficulty. As mentioned in the MS, “The individual components of Mixture 1 could not be separated further by the chromatographic systems that we used.” (Page 5, highlighted in grey). We have tried GC and HPLC methods but could not establish a linear relationship to determine the content of the individual compounds in the mixtures (This sentence has been included in the revised MS on Page 8 and highlighted in green). The high purity requirements of the various NMR techniques

and the HR-MS technique used for the successful structure elucidation might also suggest that the purity of the HOEAs compounds was very high in the mixtures. Therefore, although we cannot exclude the existing of minor impurities in the isomeric mixture, the impurities are not the major component of the toxins; their contribution to the toxicity was not big enough to affect our determination of the toxicity of the isomeric mixture.

The separation of these isomers to obtain the pure individual compounds is extremely difficult, due to possible Z/E isomerization during the separation process (12-hydroxy-10E-octadecenoic acid and 12-hydroxy-10Z-octadecenoic acid are 2 of the 3 compounds in the mixture). We have tried many times to separate them using different methods but all failed. Naturally, these compounds also exist as a mixture in the seed/seed cake. Furthermore, even though we may finally find methods to separate the individuals of the isomers to some extent, it is impossible to identify them due to the absence of standards. Therefore, we gave up our efforts to separate the individuals and determine their relative contents in the mixtures.

3) The protocol for Dess-Martin oxidation of the mixture 1 is not reported anywhere in the manuscript, so the reaction cannot be replicated by the reader.

Response: We have added the protocol for Dess-Martin oxidation of the mixture 1 in the revised MS (Page 31-32, in green).

4)The toxicology experiment section is characterized by poor experimental design. For example, the doses and regimen used for the administration of the extracts in *in vivo* experiments lacks of rational explanation.

Response: In both the **Results** and the **Methods** parts, we have included some explanations as you advised which are highlighted in green. The general aim of the *in vivo* animal toxicity tests was to confirm whether HOEAs cause the same toxicity as *Jatropha* seeds/seed cake (This sentences has been included in the revised MS in Methods on Page 34). The dosages were estimated by the results of the carp fingerling. The results indicate

that the dosages and the regimes that we used were effective and efficient for the aim of the experiments.

The paragraph in which is described the incident occurred to the researcher (spillage of the extract on the skin) does not add any scientific information on the manuscript and need to be deleted.

Response: This part has been deleted.

In different part of the manuscript hypothesis are advanced by the authors with no scientific evidence of it (for example, reasons for no toxicity after HOEAs gavage feeding). This is better to be avoided.

Response: The discussion on the possible reasons for less toxicity after HOEAs gavage feeding as compared with intravenous injection has been deleted as you suggested.

As described in the MS, although HOEAs gavage feeding did not cause strong acute toxicity (immediate death) as intravenous injection, it was toxic, not non-toxic. The feeding caused diarrhea. In consistent with our results, when *Jatropha curcas* cake is used as feed, it also does not cause an immediate death of the animals, but does cause diarrhea, and cause weight loss and malnutrition, even animal death over long time continuous feeding. For a better understanding, we included "Similar to our results, feeding *KCake* also did not cause immediate death of mice²⁹, rats^{20, 26}, rabbits²⁶, calves²⁵, goats^{30, 31}, sheep^{30, 32} and pigs²³." in the revised MS (Page 16, in green).

5) The author tried to determine a structural-toxicology relationship along all the manuscript referring to single functional groups that constitute the molecules. From the paper results that having a hydroxyl group is giving a worst toxicological profile to the molecules. As it is presented, this is conceptually wrong, since is the not just the presence of a single functional group responsible for the biological activity of the molecule.

Response: Our efforts were to address the role/importance of the hydroxyl group and the double bond on the toxicity. The combination of results shown in Table 1 and the chemical

structures of the compounds shown in Fig. 2 (Fig. 1 in the previous MS) suggests the role of the hydroxyl group and double bond in the toxicity of the fatty acids. In order to further confirm the role of hydroxyl group, we included oleic acid (9Z-octadecenoic acid), an octadecenoic acid that has the same basic carbon-chain structure as the individuals of the HOEAs but does not have a hydroxyl group, as an additional control in some of the studies on the toxicity mechanism. The results shown that oleic acid did not stimulate the production of ROS (Fig. 5), a key early event of the toxicity signal pathway of HOEAs (Fig. 9), while HOEAs did. Furthermore, HOEAs were significantly stronger in promoting mast cell degranulation (the main pathological cause of allergies and urticaria) and in promoting the growth of MCF-7 breast cancer cells (Fig. 7). All these results suggest the role/importance of the hydroxyl group in the observed toxicity.

In order to remove your concern, we have modified the related expressions, from “are the function groups of” to “are essential for” in the Abstract, “determine the toxicity” to “play a key role in the toxicity” “the structure basis of the toxicity” to “is required for the toxicity”, “is responsible for” to “is required for” in Results and Discussion (all highlighted in yellow).

In addition, we included the following paragraph in the revised MS in Discussion to support the role of hydroxyl group on the toxicity (Page 29, Line 469-476, in green):

Consistent with our findings, the hydrated products of leukotoxin (Ltx, 9,10-epoxy-12Z-octadecenoic acid) and isoleukotoxin (iLtx, 12,13Z-epoxy-9Z-octadecenoic acid), leukotoxin diol and isoleukotoxin diol, are more toxic than the parent Ltx and iLtx, which suggests the role of the diols in the toxicity of dihydroxy fatty acids (Moghaddam et al., 1997). The hypothetical “diol-specific” mechanism of toxicity was further confirmed by analyzing the cytotoxicity of 75 various parent olefins with different carbon chain length and their corresponding epoxides and diols (Greene et al., 2000). ”

In conclusion the manuscript as it is presented presents several issues that need to be solved by the authors for publication

Response: We hope that our explanations and the added data have removed your concerns.

Reviewer #3 (Remarks to the Author):

Reviewing the manuscript entitled “Hydroxy-octadecenoic acids not phorbol esters are responsible for the toxicity of *Jatropha curcas* kernel cake”.

Overall recommendation: The manuscript investigate the toxic effects of phorbol esters (PEs) of *Jatropha curcas* (KCakeJ) using different animal models and cell lines. The toxic effects were well demonstrated and the presented results and the conclusion were discussed elegantly, that make it acceptable for publication. However, further additions and modifications should be considered in order to improve the manuscript quality before publication.

Response: We thank you very much for your critical review, insightful comments and helpful suggestions on how to improve our MS. We hope our answers will remove your concerns.

- Figures,

1- One of NMR figures should be moved to the main text as NMR spectroscopy is mainly used for identification and structural elucidation

Response: The DEPT ^{13}C spectrum of Mixture 1 has been add to the revised MS as part of Fig. 1 (Page 7).

2- The quality of figure 4 mainly Figure 4B and Figure 4C should be improved

Response: It has been improved.

3- The text within Figure 7 is not readable. Thus the front size should be improved

Response: It has been improved.

- In the conclusion authors should not exaggerate the finding page 27 line 449 “This work marks a revolutionary advance”. Instead it is better to say the results..suggest , propose, show

Response: That sentence has been removed. Similar expressions have been modified as you advised (highlighted in yellow).

- The samples scale of the study is very large (Kg of samples Litters of solvents) which is not usually of laboratory studies where sample size is usually mg and ml of solvents. Authors should comment in this and explain why they used such large volume of sample and how sure they are from the purity of the solvent with this large solvent. Is it possible that solvent impurities may concentrated and interfere with the results?

Response: We have included the following paragraphs in the revised MS to answer your questions (the first 2 paragraphs on Page 30-31, highlighted in green; the last paragraph on Page 5, highlighted in green).

In the pre-experiments, we found that silica absorbed a big quantity of the sample during the chromatography progress. Every column chromatography process resulted in a big loss of the sample. Due to the difficulty to separate the toxins, the separation process included many repeated chromatography processes. Therefore, the whole separation process would lose a big quantity of sample. In addition, we hoped to obtain enough major toxins for future studies, such as, to do the toxicity tests and to do the biotransformation experiments for the detoxification studies. Furthermore, we wanted to check whether it is possible to obtain PEs from an extremely big sample size. Therefore, we started the extraction and isolation process with 2,000 kg KCakeJ.

The technical grade solvents used in the initial extraction and separation processes, including methanol (obtained from Yunnan Yun Feng Chemical Engineering Ltd), acetyl acetate (obtained from Guangxi Yi Tian Chemical Engineering Ltd) and petroleum ether (b.p. 60-90 °C, obtained from Shandong Tian De Chemical Engineering Ltd), were distilled under 0.06 MPa to remove the impurities prior to use. During the distillation process, the top 10% and the bottom 10% of the distilled solvents were discarded; the middle 80% was collected for use. Solvents less than 10 liters were distilled in a vacuum rotary evaporator; bigger quantity of solvents were distilled in a 10 m³ steel distillation equipment. Methanol, acetyl acetate and petroleum ether were distilled at 50 °C, 42 °C and 35-40 °C, respectively.

The bulk technical grade solvents used in the extraction and separation processes were purified by distillation prior to use. Because the solvents used have a much lower boiling point than those of the individuals of Mixture 1, the Mixture 1 should have been removed from the solvents during the distillation process prior to use even though they had existed in the original commercial solvents; All the other solvents used were analytical reagents. In addition, after we obtained Mixture 1 from KCakeJ, we also obtained Mixture 1 by extraction from Jatropha seed oil in some independent experiments (data not shown). The extraction process from oil contained much less steps than that from kernel cake and were conducted at normal laboratory scale using analytical reagents throughout the whole process. Therefore, we exclude the possibility that Mixture 1 were from the solvents.

- Is the detected compounds are commercially available, is it possible to used standard samples of these compounds to compare the NMR spectra of standard sample with those separated from *Jatropha curcas*

Response: It is a great idea to make the comparison as you suggested. However, we believed that to make the comparison has technical difficulty for the following reasons: These compounds are not commercially available. In addition, we have tried many times but could not separate the individuals by HPLC and other chromatography methods. Thus the content of the individual hydroxyl fatty acid in the isomeric mixtures is still unknown.

Finally this work is import and presents a valuable information that may help to understand the toxic roles of isomeric mixture of 12-hydroxy-10E-octadecenoic acid, 11-hydroxy-9E-octadecenoic acid and 12-hydroxy-10Z-octadecenoic acid (HOEAs)

Response: Thanks a lot for your kind recommendation for our paper.

Once again thanks a lot to all the reviewers for your time and attention.

Yours sincerely,

Changhe Zhang, representing all the authors

References added:

Greene, J. F., Newman, J. W., Williamson, K. C., and Hammock, B. D. (2000). Toxicity of epoxy fatty acids and related compounds to cells expressing human soluble epoxide hydrolase. Chem. Res. Toxicol. 13, 217-226.

Moghaddam, M. F., Grant, D. F., Cheek, J. M., Greene, J. F., Williamson, K. C., and Hammock, B. D. (1997). Bioactivation of leukotoxins to their toxic diols by epoxide hydrolase. Nat. Med. 3, 562-566.

REVIEWERS' COMMENTS:

Reviewer #1 (Remarks to the Author):

In the revised manuscript, authors responded my major concerns. However, the quality of Fig.5 and Fig.6 should be improved.

Reviewer #2 (Remarks to the Author):

The author offered an improved version of the manuscript after the comments of the referees. This included a precise explanation of the structural determination process for the isolated compounds. Although this clearly improve the quality of the manuscript, the lack of methods for the quantitative determination of single compounds in the mixture is still a major conceptual flaw. In fact, this prevent a precise characterization of the toxicology features of the isolated fractions. For this reason the conclusions reported in the manuscript, in the actual form, do not result convincing and fully sustained by evidences.

Response to the Reviewers

REVIEWERS' COMMENTS:

Reviewer #1 (Remarks to the Author):

In the revised manuscript, authors responded my major concerns. However, the quality of Fig.5 and Fig.6 should be improved.

Response: Thanks a lot for your comments. The graphics have been modified with high resolution statistics box-whisker plots. We exchanged the position of Fig. 5a and Fig.6, which was wrongly placed in the previous version, as well.

Reviewer #2 (Remarks to the Author):

The author offered an improved version of the manuscript after the comments of the referees. This included a precise explanation of the structural determination process for the isolated compounds. Although this clearly improve the quality of the manuscript, the lack of methods for the quantitative determination of single compounds in the mixture is still a major conceptual flaw. In fact, this prevent a precise characterization of the toxicology features of the isolated fractions. For this reason the conclusions reported in the manuscript, in the actual form, do not result convincing and fully sustained by evidences.

Response: Thanks a lot for your comments. We are sorry for not being able to explain the issue clearly to remove your concerns in our previous response to you. We have included a paragraph at the end of Results to tone down the conclusion to remove your concerns.

It is ideal to quantify the relative percentage of each of the 3 isomers of hydroxy-octadecenoic acids. Due to lack of commercial standards of the individual isomers of the hydroxy-octadecenoic acids we were not able to establish a linear relationship to determine the content of the individual isomers by GC and HPLC methods. However, the high sample purity requirements ($\geq 95\%$) of the various NMR techniques and the HR-MS technique used for the successful structure elucidation of the isomers and the fact that mixture 1 could not be separated by the chromatographic methods as mentioned previously suggest that the purity of the isomers in Mixture 1 was very high. Therefore, although we cannot exclude the existence of minor unknown impurities in hydroxy-octadecenoic acids, the impurities

were not the major component of the toxins and their contribution to the toxicity was not big enough to affect our determination of the toxicity of the isomeric mixture if they did exist. The determination of the toxicity of each individual isomer was impossible due to that they are neither commercially available nor possible to be separated from hydroxy-octadecenoic acids. Hydroxy-octadecenoic acids are from Jatropha seed. Naturally, the 3 individual isomers of hydroxy-octadecenoic acids co-exist in the seed and kernel cake, and also exert toxicity together. The determination of the toxicity of hydroxy-octadecenoic acids on the whole mimics their natural toxicity characteristics in seed and kernel cake. Therefore, for the determination of the toxicity of hydroxy-octadecenoic acids it is not necessary to quantify the relative percentage of each of the 3 isomers.